# WhyD tailors surface polymers to prevent premature bacteriolysis and direct cell elongation in *Streptococcus pneumoniae*

Josué Flores-Kim[1,2]*[†], Genevieve S Dobihal[1][†], Thomas G Bernhardt[1,3]*, David Z Rudner[1]*

[1]Department of Microbiology, Harvard Medical School, Boston, United States; [2]UMass Chan Medical School, Worcester, United States; [3]Howard Hughes Medical Institute, Boston, United States

**\*For correspondence:**
Josue.FloresKim3@umassmed.edu (JF-K);
thomas_bernhardt@hms.harvard.edu (TGB);
rudner@hms.harvard.edu (DZR)

[†]These authors contributed equally to this work

**Competing interest:** The authors declare that no competing interests exist.

**Abstract** Penicillin and related antibiotics disrupt cell wall synthesis in bacteria causing the downstream misactivation of cell wall hydrolases called autolysins to induce cell lysis. Despite the clinical importance of this phenomenon, little is known about the factors that control autolysins and how penicillins subvert this regulation to kill cells. In the pathogen *Streptococcus pneumoniae* (*Sp*), LytA is the major autolysin responsible for penicillin-induced bacteriolysis. We recently discovered that penicillin treatment of *Sp* causes a dramatic shift in surface polymer biogenesis in which cell wall-anchored teichoic acids (WTAs) increase in abundance at the expense of lipid-linked teichoic acids (LTAs). Because LytA binds to both species of teichoic acids, this change recruits the enzyme to its substrate where it cleaves the cell wall and elicits lysis. In this report, we identify WhyD (SPD_0880) as a new factor that controls the level of WTAs in *Sp* cells to prevent LytA misactivation and lysis during exponential growth . We show that WhyD is a WTA hydrolase that restricts the WTA content of the wall to areas adjacent to active peptidoglycan (PG) synthesis. Our results support a model in which the WTA tailoring activity of WhyD during exponential growth directs PG remodeling activity required for proper cell elongation in addition to preventing autolysis by LytA.

## Editor's evaluation

The mechanisms by which cell wall hydrolases are controlled to prevent their lethal misactivation are not well understood. This study reports the identification and characterization of an enzyme, named WhyD, that specifically hydrolyzes wall teichoic acids (TAs) in Streptococcus pneumoniae. Importantly, WhyD activity depletes TA content at active PG synthesis sites. Because the major autolysin LytA binds to TAs, this mechanism prevents its action at these sites.

## Introduction

Most bacterial cells are surrounded by a cell wall matrix composed of peptidoglycan (PG). This exoskeletal layer fortifies the cell membrane against internal osmotic pressure and is essential for cell integrity. The PG consists of glycan polymers with a repeating disaccharide unit of N-acetylglucosamine (GlcNAc) and N-acetylmuramic acid (MurNAc). Attached to the MurNAc sugar is a short peptide that is used to form crosslinks between adjacent glycan polymers, generating an interconnected PG matrix. Synthesis of the PG heteropolymer is mediated by the penicillin-binding proteins (PBPs), some of which possess both glycosyltransferase and transpeptidase activity needed to polymerize and crosslink the glycan strands of PG, respectively (*Goffin and Ghuysen, 1998*; *Sauvage et al., 2008*). The glycans are also polymerized by SEDS (shape, elongation, division and sporulation)-family

proteins that work in unison with monofunctional PBPs possessing crosslinking activity (*Cho et al., 2016*; *Emami et al., 2017*; *Meeske et al., 2016*; *Rohs et al., 2018*; *Sjodt et al., 2018*; *Taguchi et al., 2019*). Because PBPs are the targets of penicillin and related β-lactam drugs (*Cho et al., 2014*; *Strominger and Tipper, 1965*; *Tipper and Strominger, 1965*), much of the research focus in the field of cell wall biology has been on the regulation of cell wall synthesis by these enzymes and their inhibition by cell wall targeting antibiotics.

Since the PG layer is a continuous matrix, cell wall biogenesis also requires the activity of enzymes that cut bonds in the network called PG hydrolases or autolysins. These space-making factors are important for breaking bonds in the matrix to allow for its expansion and the insertion of new material into the pre-existing meshwork (*Bisicchia et al., 2007*; *Carballido-López et al., 2006*; *Dobihal et al., 2019*; *Dörr et al., 2013*; *Meisner et al., 2013*; *Singh et al., 2012*; *Sycuro et al., 2010*; *Vollmer et al., 2008*; *Wilson and Garner, 2021*). They also play important roles in cleaving shared cell wall material connecting daughter cells during cytokinesis (*Hashimoto et al., 2018*; *Heidrich et al., 2002*; *Vollmer et al., 2008*; *Wilson and Garner, 2021*). Given their potential to induce cell wall damage, it has long been appreciated that bacteria must employ robust mechanisms to prevent aberrant PG cleavage and lysis by autolysins. Notably, β-lactams and related antibiotics have long been known to kill bacteria by corrupting the activity of PG hydrolases to damage the cell wall and cause catastrophic lysis (*Cho et al., 2014*; *Salamaga et al., 2021*; *Tomasz and Waks, 1975*; *Tomasz et al., 1970*). Despite the relevance of this phenomenon for antibiotic development, surprisingly little is known about the regulatory mechanisms governing when and where autolysins are activated during normal growth and how antibiotics disrupt these processes to induce lysis. Identifying the factors involved would address an outstanding question in microbiology and reveal attractive new vulnerabilities in bacterial cells to target for the discovery of novel lysis-inducing drugs.

One of the main challenges in elucidating mechanisms controling PG hydrolase activity has been the genetic redundancy of these enzymes. Mutants defective in one or even several PG hydrolases rarely display a phenotype that can be exploited for genetic analysis of their regulation. To circumvent this difficulty, we used the gram-positive pathogen *Streptococcus pneumoniae* (*Sp*) because it only requires a single autolysin, LytA, to trigger its characteristic growth-phase-dependent and antibiotic-induced lysis phenotypes (*Tomasz and Waks, 1975*; *Tomasz et al., 1970*). This reliance on a single enzyme for lysis induction allowed us to design a transposon sequencing (Tn-Seq) screen for factors that control LytA activity based on the identification of essential genes that become dispensable when LytA is inactivated. The screen revealed that LytA regulation is intimately linked to the biogenesis of surface glycopolymers called teichoic acids (TAs) (*Flores-Kim et al., 2019*).

TAs are major constituents of the gram-positive cell envelope and include both lipoteichoic acids (LTAs), which are membrane anchored, and wall teichoic acids (WTAs), which are covalently attached to the PG (*Brown et al., 2013*). In *Sp* cells, both types of TAs are made from a common precursor linked to the lipid carrier undecaprenyl phosphate (Und-P) (*Brown et al., 2013*; *Denapaite et al., 2012*; *Fischer et al., 1993*; *Gisch et al., 2013*; *Heß et al., 2017*). They are also decorated with choline moieties, which serve as binding sites for LytA and other *Sp* enzymes with choline-binding domains (CBDs) (*Brown et al., 2013*; *Denapaite et al., 2012*; *Fischer et al., 1993*; *Gisch et al., 2013*; *Heß et al., 2017*). We previously found that during exponential growth, LTAs predominate and sequester LytA at the membrane thereby preventing cell wall damage (*Flores-Kim et al., 2019*). However, upon antibiotic treatment or upon prolonged periods in stationary phase, the membrane protease FtsH degrades the LTA synthase TacL, causing a dramatic decrease in LTA levels, and an increase in WTAs. This switch in TA polymer abundance leads to the relocalization of LytA from the membrane to the cell wall where its cleavage activity compromises wall integrity, inducing lysis (*Flores-Kim et al., 2019*).

In this report, we characterized another essential *Sp* gene, *spd_0880*, that becomes dispensable in cells lacking LytA. We have renamed this gene *whyD* (WTA hydrolase) based on our results showing that it encodes a membrane-anchored enzyme that removes WTAs from the *Sp* cell wall. Cells inactivated for WhyD accumulate high levels of WTAs in their cell wall during growth when these polymers are normally low in abundance relative to LTAs. Thus, LytA is constitutively recruited to the wall in *whyD* mutant cells where its PG cleavage activity elicits lysis. Notably, we found that WhyD is not only needed to control the abundance of WTAs but is also required to restrict their localization in the wall to areas of cell elongation. We further show that the activity of choline-binding PG hydrolases that associate with WTAs is important for the normal elongation of *Sp* cells. Taken together, our results

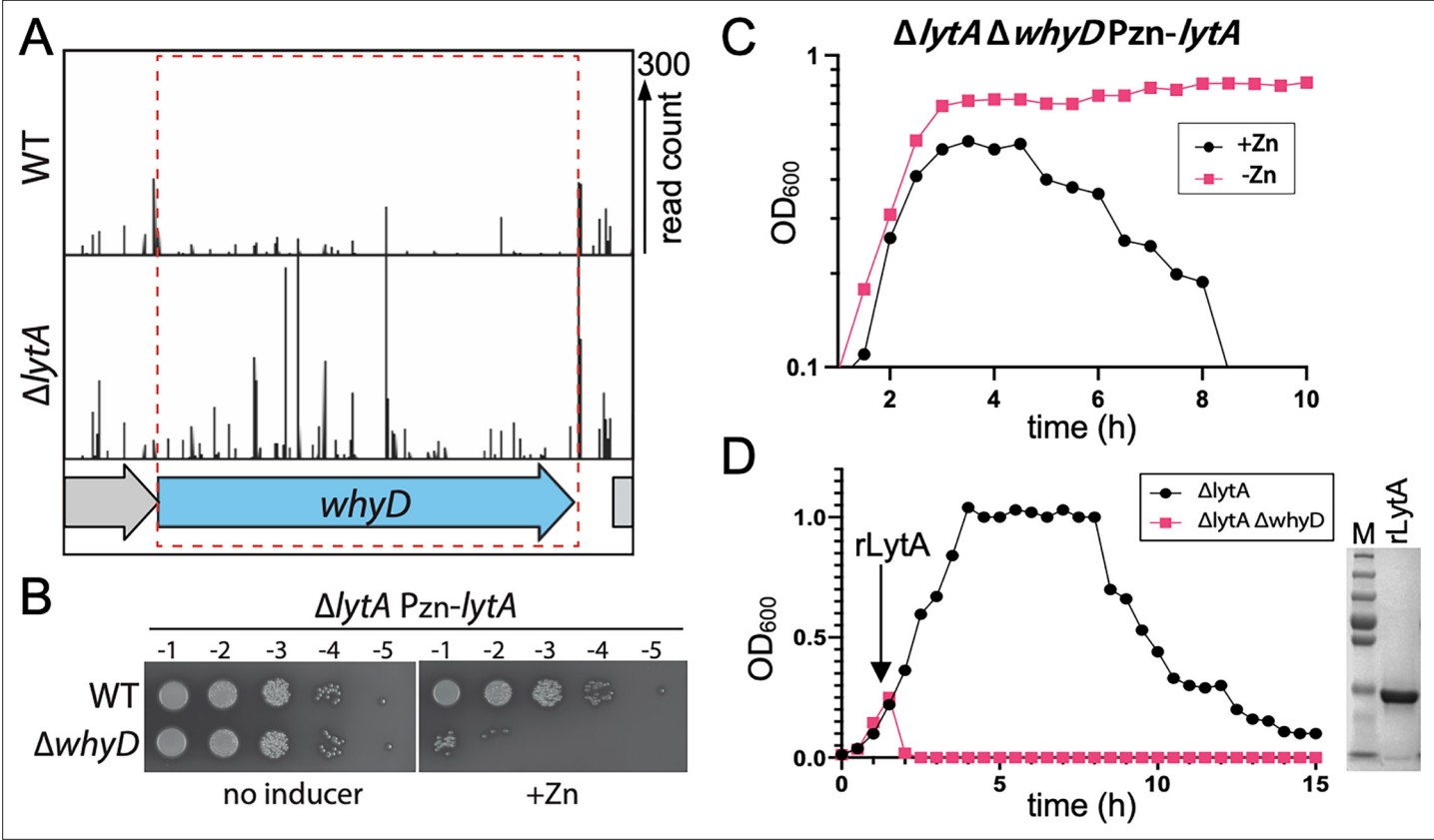

**Figure 1.** *WhyD* essentiality is *lytA*-dependent. (**A**) Transposon insertion profiles in wild-type (WT) and the Δ*lytA* mutant. Separate Mariner transposon libraries were generated, and insertion sites and their abundance were mapped to the *Streptococcus pneumoniae* genome. A region including the *whyD* locus is shown. Each line indicates an insertion site, and its height reflects the number of sequencing reads. Transposon insertions in *whyD* were under-represented in WT compared to Δ*lytA*. (**B**) Serial dilutions of the indicated strains in the presence (+Zn) and absence of inducer. Cells were grown to exponential phase, normalized and tenfold serially diluted. Aliquots (5 µl) of each dilution were spotted onto TSAII 5% SB plates in the presence or absence of 200 µM $ZnCl_2$. Plates were incubated at 37 °C in 5% $CO_2$ and then imaged. (**C**) Expression of *lytA* in cells lacking *whyD* results in growth arrest and lysis in exponential phase. Strains containing a zinc-inducible *lytA* allele (Pzn-*lytA*) were grown in THY to mid-exponential phase. Cultures were diluted into fresh THY to an $OD_{600}$ of 0.025 in the presence or absence of 200 µM $ZnCl_2$ and grown at 37 °C in 5% $CO_2$. Growth was monitored by $OD_{600}$ measurements approximately every 30 min for 10 hr. (**D**) Cells lacking whyD are sensitive to exogenous LytA. Growth curves of the indicated strains before and after the addition of 1 µg/ml recombinant LytA (rLytA) at an $OD_{600}$ of ~0.2. Growth was monitored by $OD_{600}$ approximately every 30 min for 15 hr. The Δ*whyD* mutant rapidly lysed after rLytA addition. By contrast and as reported previously, the Δ*lytA* strain lysed in stationary phase in a manner similar to LytA⁺ cells. Right: Coomassie-stained gel of rLytA purified from *Escherichia coli* (*E. coli*). Molecular weight markers (**M**) are shown.

The online version of this article includes the following source data for figure 1:

**Source data 1.** Serial dilutions of WT and Δ*whyD* strains harboring an inducible copy of *lytA* in the presence (+Zn) and absence of inducer (*Figure 1B*).

**Source data 2.** Coomassie-stained gel of rLytA purified from *E. coli* (*Figure 1D*).

support a model in which the tailoring of WTAs by WhyD helps direct the activity of space maker PG hydrolases to locations of cell wall expansion in addition to preventing LytA-induced autolysis during exponential growth .

# Results

## Identification of WhyD

To identify regulators of autolysis, we previously performed a Tn-Seq screen for essential *Sp* genes that become dispensable in a Δ*lytA* mutant (*Flores-Kim et al., 2019*). In addition to *tacL* described in our original report, we found that the *whyD* (*spd_0880*) gene also displayed a pattern of essentiality/non-essentiality expected for a LytA regulator (*Figure 1A*). In wild-type (WT) cells, relatively few transposon insertions were mapped in *whyD*, consistent with previous genomic studies that reported

it to be an essential gene (*Liu et al., 2017*; *Liu et al., 2021*; *van Opijnen and Camilli, 2012*). By contrast, in cells lacking LytA, insertions in *whyD* were readily detected (*Figure 1A*). To validate the Tn-seq results, we constructed a LytA-depletion strain in which the sole copy of *lytA* was under the control of a zinc-regulated promoter (*Eberhardt et al., 2009*). When LytA was absent (no inducer), cells were viable in the presence or absence of *whyD* (*Figure 1B*). However, when LytA was expressed (+Zn), viability was severely compromised only in cells deleted for *whyD* (*Figure 1B*). Furthermore, LytA production in cells lacking WhyD during growth in liquid medium caused premature lysis in late exponential phase (*Figure 1C*). Finally, the sensitivity of the ΔwhyD strain to LytA activity could be recapitulated by the addition of purified recombinant LytA (rLytA) to cells (*Figure 1D*). Addition of rLytA to a ΔlytA mutant during exponential growth restored the stationary phase autolysis phenotype exhibited by WT cells whereas its addition to a ΔlytA ΔwhyD double mutant resulted in lysis almost immediately after exposure (*Figure 1D*). Thus, *whyD* has the properties expected for a gene encoding a factor that restrains LytA activity at the cell surface.

## Cells lacking WhyD contain high levels of WTAs

Given our previous findings that LytA activity in *Sp* is controlled by the balance of LTAs versus WTAs (*Flores-Kim et al., 2019*), we tested the effect of WhyD inactivation on the levels of these surface polymers in exponentially growing cells (*Figure 2*). LTAs were detected in membrane preparations by immunoblotting with commercial antibodies specific for the phosphocholine (PCho) modifications whereas WTAs were detected in a parallel set of samples by alcian blue-silver staining of polymers released from purified cell wall sacculi. As a control, we analyzed LTA and WTA levels in mutants inactivated for the LTA synthase TacL. As expected, LTAs were undetectable in these cells and WTA levels dramatically increased (*Figure 2*). In mutants defective for WhyD, a similarly dramatic increase in WTAs was observed. However, in this case, LTA levels were unaffected (*Figure 2*). Expression of *whyD* from an ectopic locus restored WT levels of WTAs, indicating that the phenotype was due to the absence of the WhyD protein rather than an effect of the deletion on the expression of a nearby gene (*Figure 2*).

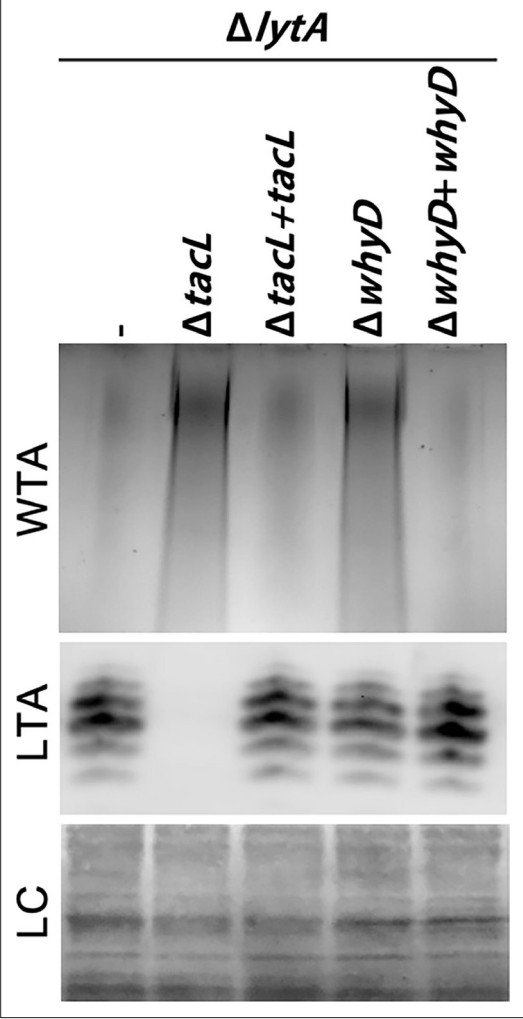

**Figure 2.** Cells lacking WhyD have elevated levels of wall teichoic acids (WTAs). Analysis of WTA and lipoteichoic acids (LTA) levels in strains lacking WhyD or TacL. The indicated strains were grown to exponential phase and harvested. WTAs were released from purified cell wall sacculi and separated by SDS-PAGE followed by alcian blue-silver staining. Membrane-associated LTAs were resolved by 16% Tris-tricine SDS-PAGE, transferred to nitrocellulose and probed with an anti-phosphocholine monoclonal antibody. A region of the nitrocellulose membrane was stained with Ponceau S to control for loading (LC). Strains with complementing alleles of *whyD* or *tacL* under control of the Zn-inducible promoter were grown in the presence of 200 μM ZnCl₂.

The online version of this article includes the following source data and figure supplement(s) for figure 2:

**Source data 1.** Alcian blue-silver staining of SDS-PAGE separated WTAs, and an anti-phosphocholine blot for SDS-PAGE separated LTAs.

**Figure supplement 1.** WhyD levels remain constant throughout growth and autolytic conditions.

**Figure supplement 1—source data 1.** Immunoblot

*Figure 2 continued*

analysis of WhyD and TacL-FLAG over the growth curve.

**Figure supplement 2.** Cells lacking WhyD do not accumulate lipoteichoic acids (LTAs) under autolytic conditions.

**Figure supplement 2—source data 1.** Alcian blue-silver staining of SDS-PAGE separated WTAs and anti-phosphocholine blots for SDS-PAGE separated LTAs collected over the growth curve and post antibiotic treatment.

We previously showed that in cells treated with penicillin or those grown for an extended period in the stationary phase, TacL is degraded, leading to a decrease in LTAs and an increase in WTAs (*Flores-Kim et al., 2019*). However, unlike TacL, WhyD protein levels remained unchanged during exponential growth and under autolytic conditions (*Figure 2—figure supplement 1*), indicating that the rise in WTA abundance in the stationary phase or following penicillin treatment does not involve the degradation of WhyD. Additionally, the change in WTA/LTA abundance under autolytic conditions was found to be unaffected by WhyD inactivation (*Figure 2—figure supplement 2*). Although further work will be required to determine how WhyD activity is inhibited during the induction of autolysis, the results thus far clearly indicate that WhyD is required to prevent LytA-induced autolysis during exponential growth by limiting the accumulation of WTAs in the cell wall.

## WhyD is a WTA hydrolase

The *whyD* gene encodes a protein with seven predicted N-terminal transmembrane segments followed by an extracellular GlpQ phosphodiesterase domain (*Figure 3A* and *Figure 3—figure supplement 1*).

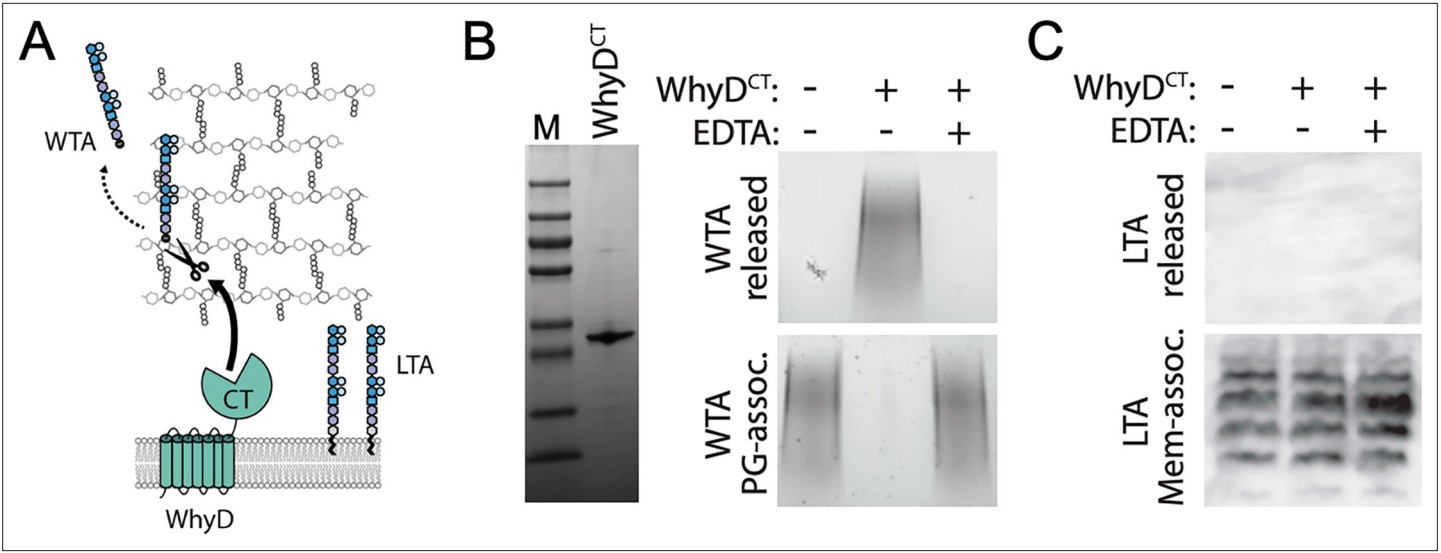

**Figure 3.** WhyD releases wall teichoic acids (WTAs) from the pneumococcal cell wall. (**A**) Schematic diagram of WhyD-dependent release of WTAs from the cell wall of *Streptococcus pneumoniae*. WTAs are attached to the MurNAc sugars of the cell wall peptidoglycan (PG) (grey) and lipoteichoic acids (LTAs) are anchored in the lipid bilayer. Teichoic acid (TA) polymers have the same chemical structure but differ in their linkages to PG or glycolipid anchor. WhyD is a polytopic membrane protein with a predicted C-terminal (CT) WTA-hydrolase domain. (**B**) The CT domain of WhyD releases WTAs from purified sacculi. Coomassie-stained gel of the recombinant CT domain of WhyD (WhyD^CT) purified from *Escherichia coli*. M, Molecular weight markers. Alcian blue-silver stained gel of WTAs released from purified sacculi (top) and those that remain associated with the PG (bottom) after incubation with 10 µg/ml WhyD^CT, 10 µg/ml WhyD^CT +1 mM EDTA, or no WhyD^CT. The reactions were incubated overnight at room temperature and then quenched with 1 mM EDTA and released WTAs collected after centrifugation. To release remaining WTAs associated with the sacculi, the sacculi pellets were treated with 0.1 M NaOH overnight at room temperature. The alkaline-hydrolyzed WTAs were then collected from the supernatant. (**C**) Immunoblot analysis of membrane preparations treated with WhyD^CT to assess its ability to release LTAs. Membranes from ΔlytA ΔwhyD cells were treated with 10 µg/ml WhyD^CT, 10 µg/ml WhyD^CT +1 mM EDTA, or no WhyD^CT. The reactions were incubated overnight at room temperature. Released and membrane-associated material were then resolved by 16% Tris-tricine SDS-PAGE, transferred to nitrocellulose, and probed with an anti-phophsocholine monoclonal antibody.

The online version of this article includes the following source data and figure supplement(s) for figure 3:

**Source data 1.** Coomassie-stained gel of WhyD^CT and alcian blue-silver stained gel of WTAs released or remaining with purified sacculi after treatment with WhyD^CT (*Figure 3B*); anti-phosphocholine blots on membrane preparations treated with WhyD^CT (*Figure 3C*).

**Figure supplement 1.** Multisequence alignment of representative GlpQ sequences.

Proteins with this domain from *Bacillus subtilis* and *Staphylococcus aureus* have recently been shown to hydrolyze WTAs (*Jorge et al., 2018*; *Myers et al., 2016*; *Walter et al., 2020*). Together with the findings presented above, we hypothesized that WhyD hydrolyzes and releases WTAs from the cell wall during exponential growth to prevent LytA recruitment to the wall and the subsequent destruction of the PG layer (*Figure 3A*). To test this possibility and to facilitate purification, we expressed and purified the soluble C-terminal GlpQ domain of WhyD (WhyD$^{CT}$; *Figure 3B* and *Figure 3—figure supplement 1*), and monitored its ability to release WTAs from purified sacculi (*Figure 3B*). Sacculi were incubated with or without purified WhyD$^{CT}$, and free WTAs in the supernatant were analyzed after pelleting. WTA polymers that remained associated with the PG sacculi in the pellet fraction were also measured following alkaline hydrolysis. As anticipated, WhyD$^{CT}$, but not buffer alone, released WTAs from sacculi into the supernatant (*Figure 3B*). Orthologues of WhyD$^{CT}$ (GlpQ) in *B. subtilis* and *S. aureus* were previously shown to require Ca$^{2+}$ ions for activity (*Jorge et al., 2018*; *Myers et al., 2016*; *Walter et al., 2020*). Similarly, we found that WTA release by WhyD$^{CT}$ was inhibited by the addition of the chelator EDTA. Given that the polymeric chemical units of WTAs and LTAs are identical in *Sp* cells, we tested whether purified WhyD$^{CT}$ was active against LTAs in membrane preparations (*Figure 3C*). Consistent with our in vivo data showing that inactivation of WhyD has no effect on the abundance of LTAs (*Figure 2*), WhyD$^{CT}$ was unable to release LTAs from purified membranes (*Figure 3C*). Altogether, these results indicate that WhyD functions as a WTA hydrolase in *Sp* cells.

## Changes in WTA levels affect cell elongation

Mutants defective for WTA biogenesis have been studied in several gram-positive bacteria, and their phenotypes have implicated these polymers in many physiological processes, including cell shape determination, cell division, virulence, and phage infection (*Boylan et al., 1972*; *Brown et al., 2012*; *Brown et al., 2013*; *Heß et al., 2017*; *Johnsborg and Håvarstein, 2009*; *Pollack and Neuhaus, 1994*; *Xia et al., 2010*; *Xia et al., 2011*; *Ye et al., 2018*). Additionally, in some gram-positive organisms like *B. subtilis*, WTAs can account for up to 30–50% of the dry weight of the cell wall (*Brown et al., 2013*; *Ellwood, 1970*). By contrast, in *Sp* cells, we find that WTAs are kept at low levels during exponential phase via the activities of TacL and WhyD (*Flores-Kim et al., 2019*). Despite this low abundance and unlike most well-studied gram-positive bacteria, WTAs are essential in *Sp* (*Johnsborg and Håvarstein, 2009*; *Ye et al., 2018*). We therefore reasoned that although WTA levels are maintained at low levels in these cells, they must be contributing to a vital part of the cell growth process.

To gain insights into the role of WTAs during growth, we modulated the levels of these polymers in *Sp* cells by inactivating or overproducing WhyD and monitoring the effects of these changes on cell morphology (*Figure 4*). Cells lacking WhyD were found to be longer, wider, and overall larger (*Figure 4A and B* and *Figure 4—figure supplement 1*). This analysis was performed in a strain lacking LytA to prevent the autolysis of cells inactivated for WhyD. We note that *lytA* inactivation had a small impact on cell size in comparison to WT, as observed previously (*Barendt et al., 2011*; *De Las Rivas et al., 2002*; *Sanchez-Puelles et al., 1986*; *Figure 4—figure supplement 2*). Conversely, overexpression of *whyD* (*whyD$^{++}$*), which lowers WTA abundance to nearly undetectable levels (*Figure 4—figure supplement 3*), reduced cell length and overall cell size (*Figure 4C–E*). These results argue that WhyD and WTAs are important for normal cell elongation.

We hypothesized that the effect of WTAs on the process of cell elongation might be related to the activity of PG hydrolases that function as space makers for the expansion of the PG layer during growth. In this case, the low levels of WTAs maintained in exponentially growing cells would ensure that PG cleavage by LytA and other PG hydrolases with CBDs occurs at levels that promote cell elongation without causing autolysis. To test this possibility, we examined the morphology of mutants containing normal WTA levels (*Figure 5C*) but lacking LytA, LytB, and LytC (Δ*lytABC*), the three main WTA-binding hydrolases expressed during exponential growth (*Figure 5*; *Kausmally et al., 2005*; *Maestro and Sanz, 2016*). Consistent with other reports and the function of LytB as the main *Sp* cell separation PG hydrolase, Δ*lytABC* cells displayed a cell separation defect (*Figure 5A*; *De Las Rivas et al., 2002*; *Zucchini et al., 2018*). However, like cells with reduced WTA levels, the cells within the Δ*lytABC* chains were also shorter and overall smaller than WT, indicative of an elongation defect (*Figure 5B*). We therefore infer that *Sp* cells likely employ WhyD to maintain a level of WTAs in the wall that are low enough to prevent autolysis but sufficient to allow WTA-binding hydrolases to promote PG expansion and cytokinesis.

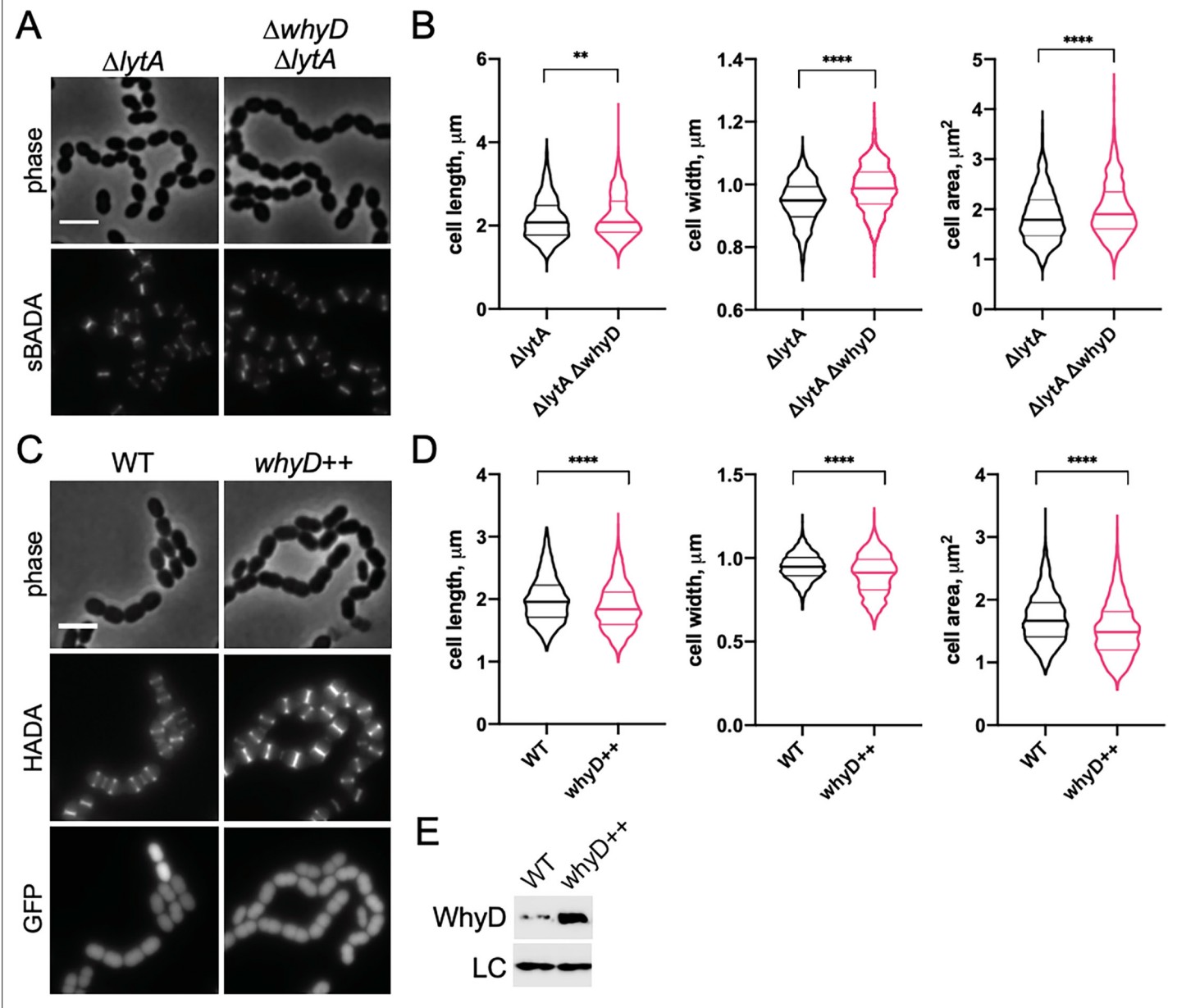

**Figure 4.** WhyD and wall teichoic acids (WTAs) are important for cell elongation. (**A**) Cells lacking WhyD are longer and larger than wild-type (WT). Representative phase-contrast and fluorescent images of Δ*lytA* and Δ*lytA* Δ*whyD* strains grown in THY at 37 °C in 5% $CO_2$ to mid-exponential phase. Cells were labeled with sBADA for 5 min prior to imaging. Scale bar, 3 μm. (**B**) Quantitative analysis of cell length, width, and area for the strains shown in (**A**). The violin plots indicate the median (bold lines) and quartiles. p-values were obtained using a Welch's t-test. p<0.01, **; p<0.0001, ****. (**C**) Cells overexpressing WhyD are shorter and smaller than WT. Representative phase-contrast and fluorescent images of WT and cells with a second copy of *whyD* under zinc-inducible control (*whyD*++). Strains were grown in THY to mid-exponential phase, diluted into fresh THY at an $OD_{600}$ of 0.025 in the presence of 200 μM $ZnCl_2$ and incubated at 37 °C with 5% $CO_2$ for 2 hr. The cells were then labeled with HADA for 5 min prior to imaging. Both strains also contain cytoplasmic GFP. Scale bar, 3 μm. (**D**) Quantitative analysis of cell length, width, and area as in (**B**). (**E**) Immunoblot analysis to assess the levels of WhyD. Samples from (**C**) were collected and normalized to an $OD_{600}$ of 0.5 and resolved by SDS-PAGE and analyzed by anti-WhyD immunoblot. LC: control for loading. WTA levels in these strains are shown in *Figure 4—figure supplement 3*.

The online version of this article includes the following source data and figure supplement(s) for figure 4:

**Source data 1.** Raw data of the quantitative analysis of cell length, width, and area (*Figure 4B*).

**Source data 2.** Raw data of the quantitative analysis of cell length, width, and area (*Figure 4D*).

**Source data 3.** Anti-WhyD immunoblotting (*Figure 4E*).

**Figure supplement 1.** Cells lacking WhyD are longer and larger than wild-type (WT).

*Figure 4 continued on next page*

*Figure 4 continued*

**Figure supplement 2.** Cells lacking LytA are similar in size to wild-type (WT).

**Figure supplement 2—source data 1.** Raw data of the quantitative analysis of cell length, width, and area.

**Figure supplement 3.** Analysis of wall teichoic acid (WTA), lipoteichoic acid (LTA), and WhyD levels in cells overexpressing WhyD.

**Figure supplement 3—source data 1.** Anti-WhyD immunoblot, alcian blue-silver stained gel of WTAs, and anti-phosphocholine immunoblot of membrane-associated LTAs.

## WhyD is enriched at sites of cell wall synthesis at midcell

To investigate whether WhyD activity is localized to specific areas within cells, we constructed several different fluorescent protein fusions. Most were not functional or resulted in unstable proteins that displayed no fluorescence. However, a fusion of GFP to the N-terminus of WhyD (GFP-WhyD) without a linker between the two proteins was fluorescent and stable enough to complement the Δ*whyD* mutant phenotype (*Figure 6—figure supplement 1A*). To investigate WhyD localization, *gfp-whyD* was expressed from a zinc-regulated promoter (Pzn-*gfp-whyD*) as the sole copy of *whyD*. Cells were grown in the presence of $Zn^{2+}$ and the fluorescent D-amino acid (FDAA) HADA to monitor both GFP-WhyD localization and active sites of PG synthesis, respectively (*Boersma et al., 2015*). Exponentially growing cells displayed a significant cytoplasmic GFP signal that was likely caused by some cleavage of the GFP-WhyD fusion (*Figure 6A* and *Figure 6—figure supplement 1B*). Nevertheless, an enrichment of GFP-WhyD at midcell was observable that co-localized with the HADA signal (*Figure 6A and B*). These data suggest that WhyD is recruited to areas of nascent PG synthesis at midcell.

## WTAs are most abundant in areas of zonal PG synthesis

We next wanted to determine whether the low steady-state level of WTAs that accumulate in *Sp* cells localizes to specific subcellular regions (*Figure 7*). To do so, we used an assay that takes advantage of LytA's ability to bind to the PCho moieties that decorate *Sp* teichoic acids (*Fernández-Tornero et al., 2001*; *Li et al., 2015*; *Mellroth et al., 2012*; *Mellroth et al., 2014*). Recombinant LytA (rLytA) and a catalytically inactive variant (rLytA*) were fluorescently labeled with Alexa-Fluor594 (*Figure 7—figure supplement 1A*; *Flores-Kim et al., 2019*; *Mellroth et al., 2012*). Importantly, rLytA-Alexa triggered growth phase-dependent autolysis at rates indistinguishable from unlabeled rLytA (*Figure 7—figure supplement 1B*), indicating that labeling did not affect LytA activity. As expected, rLytA*-Alexa did not induce lysis and was used for all imaging experiments to avoid complications of PG cleavage (*Figure 7—figure supplement 1B*). Since WTAs and LTAs are identical polymers with the same PCho moieties, we next investigated whether rLytA*-Alexa labels both polymers or exclusively WTAs. To do so, we used the $P_{zn}$-*whyD* strain that overexpresses WhyD and reduces WTA levels (*Figure 4—figure supplement 3*). Surface labeling by rLytA*-Alexa was readily detectable on WT *Sp* and cells harboring $P_{zn}$-*whyD* without exogenous $Zn^{2+}$. However, rLytA*-Alexa was undetectable when WhyD was overexpressed (+Zn) (*Figure 7—figure supplement 2*). Furthermore, we confirmed that rLytA*-Alexa exclusively labels WTAs from *Sp* (*Figure 7—figure supplement 1C*) and purified *Sp* sacculi, provided that WTAs had not been removed (*Figure 7—figure supplement 3*). Altogether, these results indicate that rLytA*-Alexa specifically binds WTAs when added to intact cells.

Having established that rLytA*-Alexa labeling can be used as a proxy for the in vivo localization of WTAs, we monitored the subcellular positions of WTAs relative to newly synthesized PG in exponentially growing cells. To follow nascent PG and recently synthesized wall material that had moved away from midcell during cell elongation, we first pulse-labeled cells with HADA and 5 min later added the compatibly labeled FDAA sBADA. We then washed the cells with medium containing 1% choline to remove native choline-binding proteins from the WTAs to ensure that the choline moieties were fully accessible to rLytA*-Alexa. Cells were then incubated with rLytA*-Alexa for 30 s, washed to remove unbound probe and imaged (*Figure 7A*). Elongating cells displayed a weak rLytA*-Alexa signal at midcell that co-localized with sBADA. The sBADA signal at midcell was flanked by two prominent fluorescent bands of rLytA*-Alexa that co-localized with HADA-labeled peripheral PG (*Figure 7B and C*). Consistent with WhyD hydrolyzing WTAs at midcell, co-localization analysis of GFP-WhyD and WTAs showed an anti-correlation between GFP-WhyD enrichment at midcell and WTA localization (*Figure 7—figure supplement 4*). In cells nearing the completion of cell division, the pattern changed with rLytA*-Alexa and HADA co-localizing within the deep cell constrictions (*Figure 7B* and

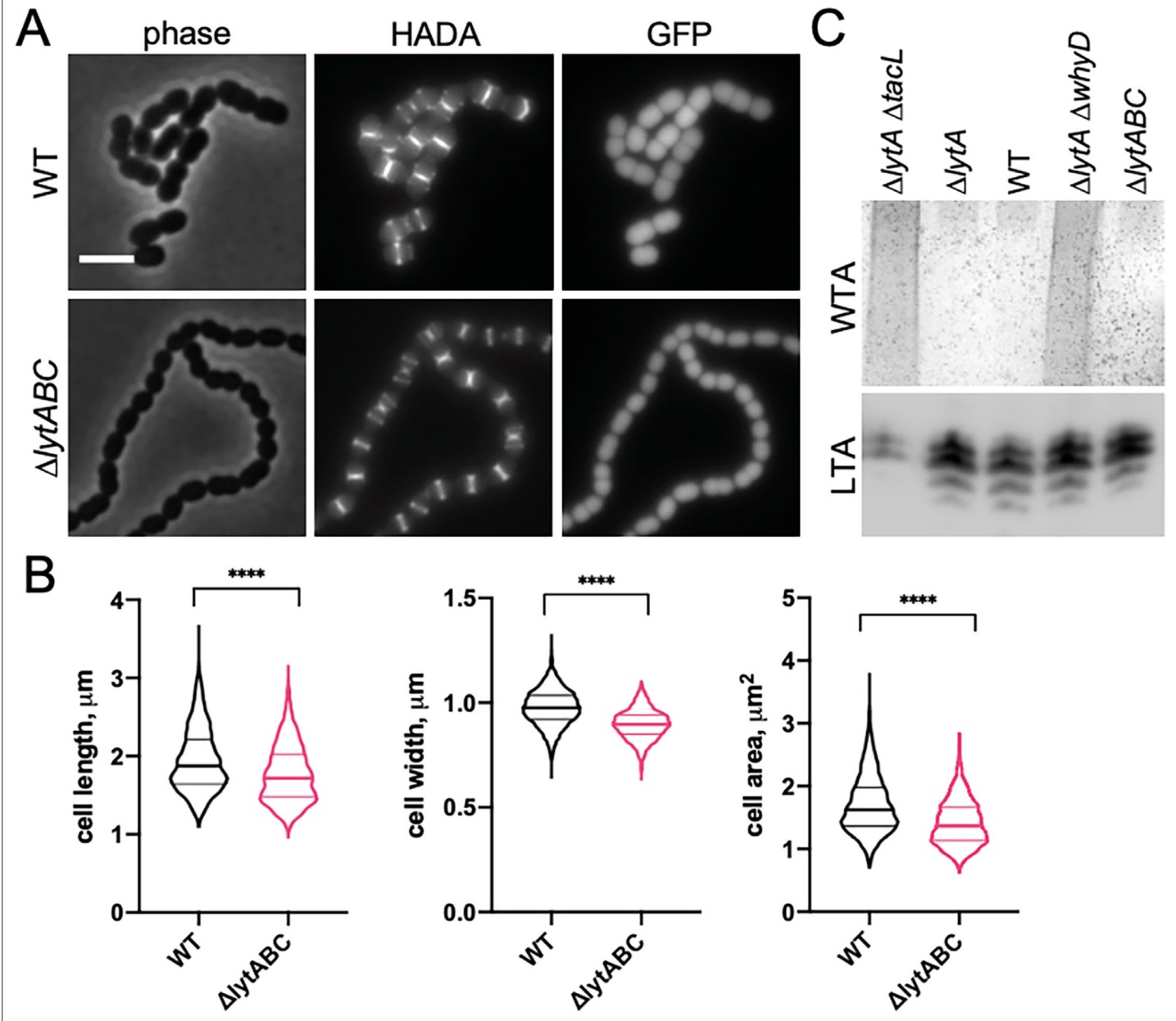

**Figure 5.** Mutants lacking wall teichoic acids (WTA)-bound cell wall hydrolases have a short cell phenotype. (**A**) Cells lacking the cell wall hydrolases that bind WTAs are shorter and smaller than wild-type (WT). Representative phase-contrast and fluorescent images of WT and cells lacking LytA, LytB, and LytC (ΔlytABC). Cells were grown in THY to mid-exponential phase and labeled with HADA for 5 min prior to imaging. Both strains also contain a cytoplasmic fluorescent marker (GFP). Scale bar, 3 µm. (**B**) Quantitative analysis of cell length, width, and area for the strains shown in (**A**). The violin plots indicate the median (bold lines) and quartiles. p-values were obtained using a Welch's t-test. p<0.0001, ****. (**C**) Analysis of WTA and LTA levels in the ΔlytABC mutant. The indicated strains were grown to mid-exponential phase and their WTA and LTA levels were analyzed as described in Materials and methods.

The online version of this article includes the following source data for figure 5:

**Source data 1.** Raw data of the quantitative analysis of cell length, width, and area (**Figure 5B**).

**Source data 2.** Alcian blue-stain of WTAst and anti-phosphocholine immunoblot of membrane-associated LTAs in the ΔlytABC mutant (**Figure 5C**).

*Figure 7—figure supplements 5 and 6*). Structured illumination microscopy (SIM) and image deconvolution analyses revealed similar localization patterns with rLytA*-Alexa localizing in regions offset from midcell in elongating cells and at deep constrictions in cells that have nearly completed division (*Figure 7—figure supplement 5C*). In support of the idea that WhyD is responsible for promoting the

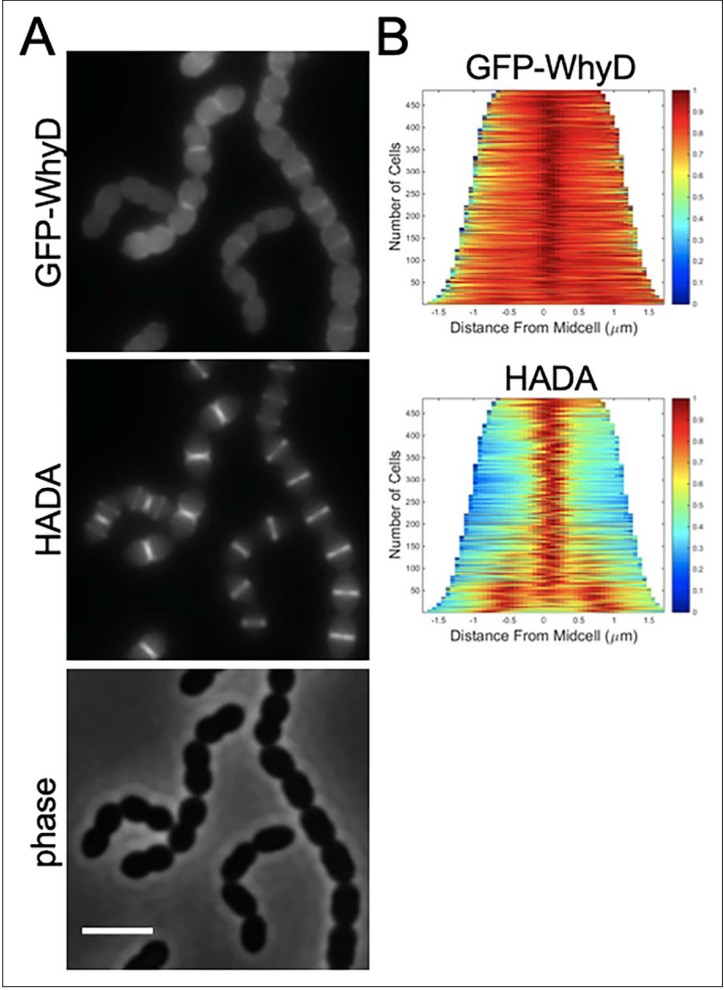

**Figure 6.** GFP-WhyD is enriched at midcell. (**A**) Representative fluorescent and phase-contrast images of cells harboring a GFP-WhyD fusion. The fusion was expressed from a zinc-inducible promoter in a Δ*whyD* background. Strains were grown in THY medium in the presence of 200 μM $ZnCl_2$ at 37 °C in 5% $CO_2$ to an $OD_{600}$ of 0.5. Cells were labeled with HADA for 5 min prior to imaging. Scale bar, 3 μm. (**B**) Demographs showing GFP-WhyD and HADA localization in a population of cells. >450 cells were quantified and the resulting heat map of fluorescence intensity for each cell was then arranged according to cell length and stacked to generate the demograph. Demographs were constructed using the open-source software package Oufti.

The online version of this article includes the following source data and figure supplement(s) for figure 6:

**Figure supplement 1.** The GFP-WhyD fusion is functional in vivo.

**Figure supplement 1—source data 1.** Serial dilutions of WT and D*lytA* strains harboring inducible copies of either *whyD* or *gfp-whyD* in the presence (+Zn) and absence of inducer.

**Figure supplement 1—source data 2.** Immunoblots of anti-WhyD and anti-GFP.

observed WTA localization, rLytA*-Alexa was present throughout the PG matrix in cells lacking WhyD (*Figure 7D*). Altogether, these results support a model (*Figure 8*) in which WhyD removes most, but not all, WTAs from nascent PG at midcell, resulting in low levels of WTAs being incorporated in the peripheral PG that brackets the septum. This localization of WTAs likely allows the recruitment of PG hydrolases to the peripheral region to promote the zonal expansion of the cell wall for elongation. Similarly, the change in WTA localization to deep constrictions in late-divisional cells is consistent with the role for the polymers in recruitment of LytB and other choline-binding PG hydrolases to catalyze the last stage of cell separation. Thus, localized pruning of WTAs from nascent PG by WhyD provides a mechanism by which *Sp* can direct the activity of its PG hydrolases to facilitate growth and division of the PG matrix (see Discussion).

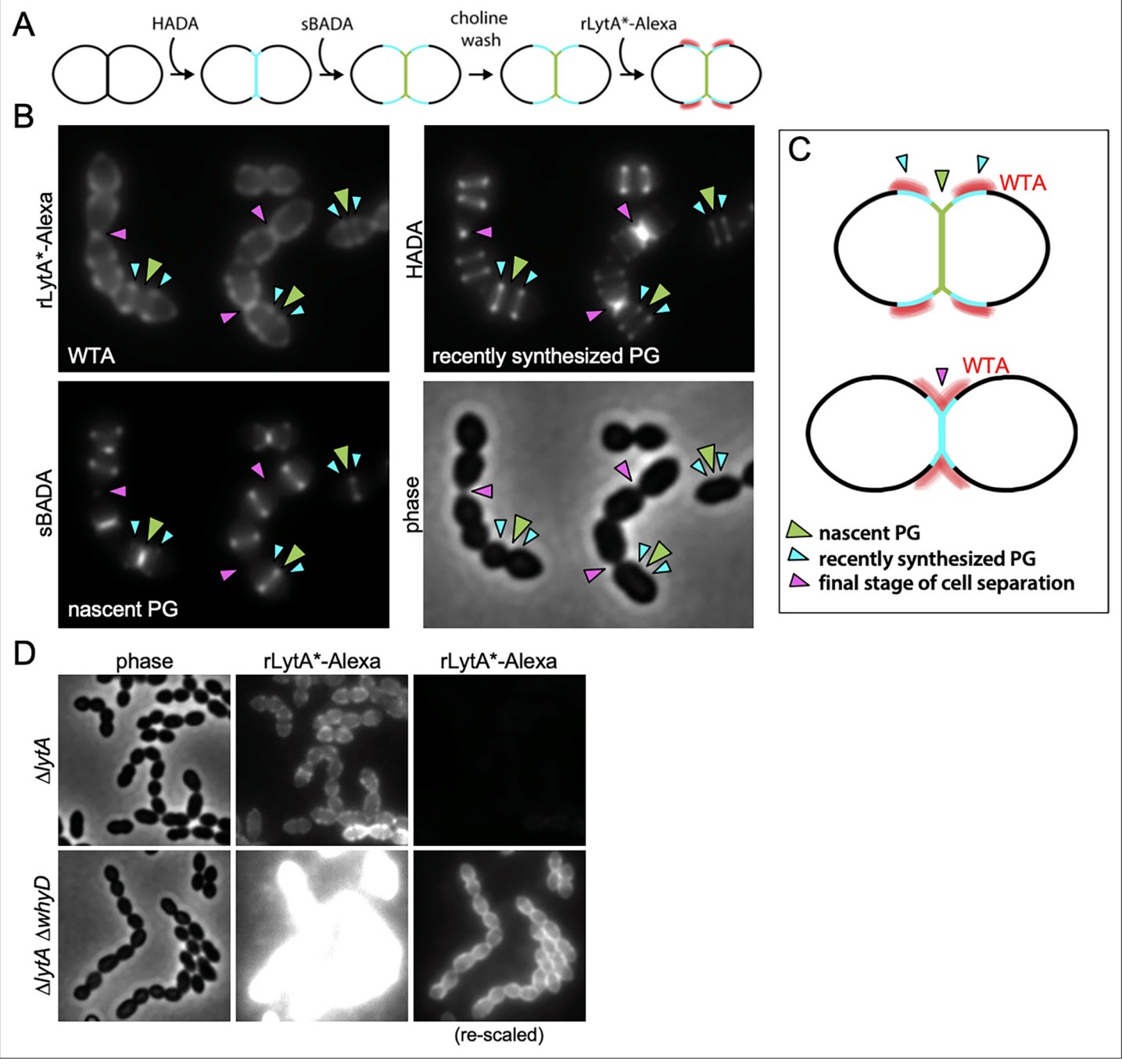

**Figure 7.** Wall teichoic acids (WTAs) accumulate adjacent to active zones of PG synthesis. (**A**) Flow diagram of experiment. The Δ*lytA* mutant was grown in THY medium to mid-exponential phase, labeled with HADA for 5 min, washed with fresh THY, and then labeled with sBADA for 5 min. The sample was then collected, normalized to an OD600 of 0.5, washed with fresh medium containing 1% choline and incubated with 1 µg/ml recombinant LytA(H26A) coupled to Alexa Fluor 594 (rLytA*-Alexa) for 30 s with gentle shaking to label WTAs. Cells were then washed twice with 1× phosphate buffer saline (PBS) and analyzed by fluorescence microscopy. (**B**) Representative phase-contrast and fluorescent images of rLytA*-Alexa (WTAs), recently synthesized (HADA) and nascent (sBADA) peptidoglycan (PG). Carets indicate nascent PG (green), recently synthesized PG (light blue), and the final stage of cell separation (pink). (**C**) Cartoon depicting WTA localization (red) relative to PG synthesis observed in (**B**). (**D**) Representative images of WTA localization in Δ*lytA* and Δ*lytA* Δ*whyD* strains. Exponentially growing cells were collected and incubated with rLytA*-Alexa as described above to label WTAs. The fluorescence intensity of the rLytA*-Alexa labeled cells was re-scaled in the right panels to visualize of WTA localization in the cells lacking WhyD.

The online version of this article includes the following source data and figure supplement(s) for figure 7:

**Figure supplement 1.** Validation of the wall teichoic acids-labeling assay.

*Figure 7 continued on next page*

*Figure 7 continued*

**Figure supplement 2.** rLytA*-Alexa specifically labels wall teichoic acids.

**Figure supplement 2—source data 1.** Raw data of the quantitative analysis of cell length, width, and area.

**Figure supplement 3.** rLytA*-Alexa specifically labels sacculi that contain wall teichoic acids (WTAs).

**Figure supplement 3—source data 1.** Alcian blue-silver stained gel of WTAs from purified sacculi.

**Figure supplement 4.** Localization of WhyD and wall teichoic acids (WTAs).

**Figure supplement 5.** Wall teichoic acids (WTAs) are enriched at the edge of zonal PG synthesis.

**Figure supplement 6.** Wall teichoic acids (WTA) localization at different stages in the cell cycle.

## Discussion

WTAs are required for normal cell growth and division in many gram-positive bacteria, and in several cases, these polymers have been implicated in controling the localization of PG hydrolases as part of their morphogenic function (*Brown et al., 2013*; *Kasahara et al., 2016*; *Schlag et al., 2010*; *Zamakhaeva et al., 2021*). However, the molecular mechanism(s) by which WTAs are localized and how they might participate in the spatio-temporal regulation of PG hydrolase activity have remained unclear for some time. Here, we show that WTA tailoring by the WhyD hydrolase plays an important role in this control process by promoting the localized accumulation of WTA polymers at sites adjacent to active areas of PG synthesis in *Sp* cells. This pruning of WTAs prevents the excessive recruitment of the LytA PG hydrolase to the cell wall to avoid autolysis during exponential growth. Additionally, our results suggest that the localization of WTAs promoted by WhyD also functions to guide the activity of WTA-binding PG hydrolases to specific subcellular sites where they can promote the remodeling of the wall necessary for proper cell elongation and division.

### WTA turnover and localization in *Sp* cells

In our previous study, we found that inactivation of the LTA synthase TacL resulted in the dramatic accumulation of WTAs in the cell wall of *Sp* cells (*Flores-Kim et al., 2019*). Because LTAs and WTAs are made from the same precursor (*Brown et al., 2013*; *Denapaite et al., 2012*; *Fischer et al., 1993*; *Gisch et al., 2013*; *Heß et al., 2017*), this observation suggested that LTAs predominate in the enve-lope in exponentially growing cells due to TacL outcompeting the WTA ligases (LCP proteins) for their common substrate. However, the discovery that WhyD inactivation also causes a dramatic increase in WTA accumulation in exponentially growing cells without affecting LTA accumulation (*Figure 2*) indicates that instead of substrate competition, it is likely that the continuous degradation of WTAs maintains their low levels in the cell wall of actively growing cells.

In addition to reducing the total WTA content attached to the PG matrix (*Figures 2 and 3*), the WTA cleavage activity of WhyD also results in the localized accumulation of these polymers at sites adjacent to areas of active wall growth (*Figure 7*). Determining how this localization is achieved will require further investigation, but this phenomenon is likely to arise from the enrichment of WhyD at midcell where most (*Figure 6*), if not all, of the nascent PG synthesis takes place in *Sp* cells (*Boersma et al., 2015*; *Briggs et al., 2021*; *Perez et al., 2021*; *Trouve et al., 2021*). Biochemical studies suggest that WTAs are most efficiently attached to nascent PG before the newly synthesized glycans are crosslinked into the pre-existing meshwork (*Rausch et al., 2019*; *Schaefer et al., 2017*). There-fore, the balance between WTA addition and cleavage at midcell could explain the observed pattern of WTA localization. In this scenario, the enrichment of WhyD in the septal region is likely to result in the removal of most but not all WTAs added to nascent PG. Zonal PG synthesis would then be expected to push the WTA-decorated PG material away from the cell center (*Figure 8*). If processing of WTAs from this older material were less efficient due to the lower concentration of WhyD outside midcell and/or the reduced accessibility of WTAs attached to more mature PG, the expected result would be a gradient of WTA accumulation centered at positions adjacent to midcell, as observed (*Figure 8*). Re-localization of WhyD to the future daughter cell septa to prepare for the next cell cycle could then be responsible for the midcell accumulation of WTAs displayed by cells in the final stages of division (*Figure 8*).

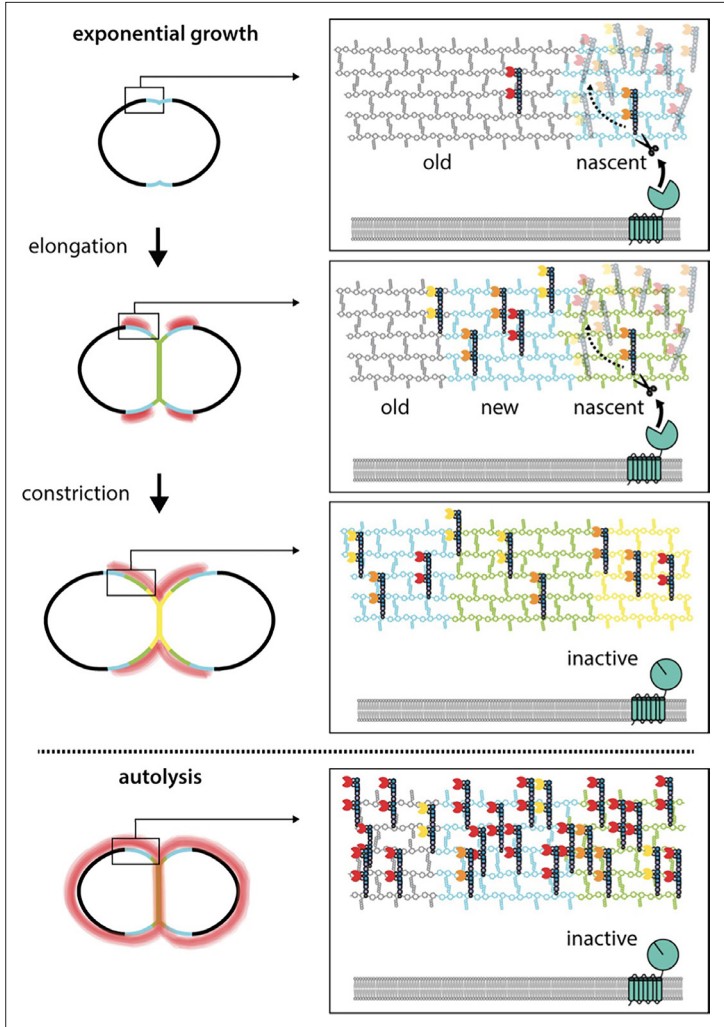

**Figure 8.** WhyD tailors wall teichoic acids (WTAs) to direct cell elongation. Schematic model of WhyD function. WhyD releases the majority WTAs attached to the cell wall during nascent PG synthesis at midcell. A subset of the WTAs remain intact and as the cell elongates these polymers recruit PG hydrolases with choline-binding domains (yellow, orange, red Pac-Men) to the zone of peripheral PG synthesis, promoting expansion of the cell wall meshwork and cell elongation. At a late stage of cell constriction, WTAs accumulate at midcell (yellow) where they recruit PG hydrolases that promote cell separation. At this stage, WhyD might not be localized at midcell or its activity could be inhibited. Upon entry into stationary phase or exposure to cell wall targeting antibiotics (autolysis), WhyD is unable to keep pace with the increase in WTA synthesis and/or is actively inhibited, leading to an increase in WTAs throughout the sacculus. Recruitment of LytA and other PG hydrolases leads to cell wall cleavage and lysis.

The online version of this article includes the following source data and figure supplement(s) for figure 8:

**Figure supplement 1.** WhyD releases TA polymers from the cell wall with a length distribution that mirrors that of LTAs.

**Figure supplement 1—source data 1.** Anti-phosphocholine immunoblot of TAs released from purified sacculi and membrane-associated LTAs.

## Possible role of WTAs in directing the activity of space-making PG hydrolases

However, WTA localization is achieved, PG hydrolases and other proteins with choline-binding domains are expected to be concentrated where the choline-containing polymers accumulate. As observed in other gram-positive bacteria, including *Streptococcus mutans*, the concentration of WTAs at late-stage septa is likely to promote the midcell recruitment of hydrolases to facilitate daughter

cell separation (*Zamakhaeva et al., 2021*). The finding that Δ*lytABC* cells lacking three major WTA-binding PG hydrolases have a short cell phenotype that resembles that of cells overproducing WhyD (*Figures 4 and 5*), which reduces their WTA content, suggests an additional role for WTA-guided PG hydrolases during cell elongation. In many bacteria, such space-making PG cleavage activity is essential for growth (*Bisicchia et al., 2007*; *Carballido-López et al., 2006*; *Dobihal et al., 2019*; *Dörr et al., 2013*; *Meisner et al., 2013*; *Singh et al., 2012*; *Sycuro et al., 2010*; *Vollmer et al., 2008*). Although WTA biogenesis is essential in *Sp* cells in accordance with such a role in cell elongation, the Δ*lytABC* mutant lacking all the known WTA-binding PG hydrolases produced during normal exponential growth is viable. Thus, either there are other WTA-binding PG hydrolases with a space-making function yet to be identified in *Sp* cells or WTAs have an essential function other than promoting cell wall expansion. In either case, the viability of the Δ*lytABC* mutant suggests that there are additional space-making PG hydrolases in *Sp* cells. Whether they are also directed by WTAs or are membrane-associated factors regulated by potential parallel pathways like PcsB and/or MltG (MpgA) remains to be determined (*Briggs et al., 2021*; *Perez et al., 2021*; *Sham et al., 2011*; *Taguchi et al., 2021*; *Trouve et al., 2021*).

## WTA cleavage activity of WhyD

WhyD has seven predicted N-terminal transmembrane segments in addition to a C-terminal GlpQ-like domain (WhyD$^{CT}$; *Figure 3A*). GlpQ-containing proteins from other gram-positive bacteria like *B. subtilis* and *S. aureus* have been shown to function as WTA hydrolases (*Jorge et al., 2018*; *Myers et al., 2016*; *Walter et al., 2020*). Unlike WhyD, which is membrane-anchored and essential for normal growth, the previously characterized GlpQ-containing proteins are secreted and function to promote growth during phosphate-limitation (*Jorge et al., 2018*; *Myers et al., 2016*; *Walter et al., 2020*). In *B. subtilis* (strain 168), LTAs and WTAs are both glycerolphosphate polymers, and *B. subtilis* GlpQ ($^{Bs}$GlpQ) has been shown to act as an exolytic enzyme that specifically degrades WTAs based on their distinct enantiomeric configuration relative to LTAs (*Brown et al., 2013*; *Jorge et al., 2018*; *Myers et al., 2016*; *Walter et al., 2020*). Our biochemical results indicate that WhyD is also specific for WTAs (*Figures 2 and 3*). However, because WTAs and LTAs in *Sp* cells are built from a common undecaprenyl-linked precursor and have an identical polymeric structure (*Denapaite et al., 2012*), WhyD is unlikely to cleave within the polymer itself. Consistent with this idea, in contrast to $^{Bs}$GlpQ, which completely hydrolyzes its WTA substrate (*Myers et al., 2016*; *Walter et al., 2020*), WhyD releases polymers from the wall with a length distribution that mirrors that of the LTAs (*Figure 8—figure supplement 1*). Given that the only major difference between WTAs in *Sp* cells relative to the precursor or the LTAs is its linkage to the PG (*Brown et al., 2013*; *Denapaite et al., 2012*; *Fischer et al., 1993*; *Gisch et al., 2013*; *Heß et al., 2017*), we favor a model in which WhyD works as an endoenzyme, specifically cleaving the phosphate linkage between the WTA polymer and PG. In support of this idea, WhyD$^{CT}$ is unable to release TA polymers from membranes (*Figure 3*), indicating that it is unable to hydrolyze LTAs or the Und-P linked TA precursor.

## Are WTAs recycled by WhyD?

A mechanism for controling the WTA content of the wall and its localization by cleaving a significant portion of the polymers that are made seems wasteful and inefficient. However, such a scenario is not that different from the synthesis of the cell wall itself, which has been reported to involve the turnover of up to 50% of the PG layer per generation in other bacteria (*Borisova et al., 2016*; *Johnson et al., 2013*). It is therefore possible that WTAs may be recycled in *Sp* cells in a manner similar to how PG turnover products are recycled in many bacterial species. A WTA recycling activity seems especially important for *Sp* given that it is auxotrophic for the choline moieties that decorate its teichoic acids. Notably, the N-terminal multi-pass transmembrane domain of WhyD shares remote homology with a family of plasma-membrane choline transporters (PF04515), raising the intriguing possibility that this domain might function to import the WTA polymers cleaved by WhyD$^{CT}$ to recycle choline and other components of the polymers.

## WhyD activity and autolysis

We discovered WhyD based on its essential function in preventing LytA-induced autolysis of *Sp* cells (*Figures 1 and 2*, *Figure 2—figure supplement 2*, and *Supplementary file 1*). Inactivation

of choline-binding hydrolase other than LytA were unable to suppress *whyD* essentiality (***Supplementary file 1***), indicating that the misactivation of other hydrolases does not play a major role in the lethal phenotype of the Δ*whyD* mutant. Like mutants lacking the LTA synthase TacL, cells inactivated for WhyD accumulate high levels of WTAs in their walls during normal exponential growth (***Figures 1 and 2***, and ***Figure 2—figure supplement 2***; ***Flores-Kim et al., 2019***). These levels of WTAs are comparable to those observed in cells treated with penicillin where the excess recruitment of LytA to the PG layer results in damage to the wall and lysis (***Figures 1 and 2***, and ***Figure 2—figure supplement 2***). Our previous work found that the accumulation of WTAs in penicillin-treated cells requires the FtsH-mediated degradation of TacL (***Flores-Kim et al., 2019***). This process presumably also requires the release of LTAs from the membrane to prevent them from sequestering LytA away from the WTAs and PG (***Flores-Kim et al., 2019***). By contrast, WhyD appears to be stable under autolytic conditions (***Figure 2—figure supplement 1***). Whether its activity must also be inhibited for autolytic induction and whether other mechanisms involved in controling autolysis exist remains unknown and requires further investigation. Nevertheless, our results clearly show that inhibition of WhyD during exponential phase has the potential to trigger cell lysis (***Figure 2—figure supplement 1***). Notably, *B. subtilis* mutants inactivated for GlpQ are hypersensitive to several β-lactam antibiotics (***Myers et al., 2016***), suggesting that WTA remodeling is generally important for controling autolysis in gram-positive bacteria. Thus, WhyD and other related GlpQ family members represent attractive targets for the development of new classes of lysis-inducing antibiotics and/or potentiators of existing β-lactam drugs.

# Materials and methods

## Key resources table

| Reagent type (species) or resource | Designation | Source or reference | Identifiers | Additional information |
|---|---|---|---|---|
| Strain, strain background (*Streptococcus pneumoniae [Sp]*) | WT (*Sp* D39 Δ*cps*) | *Lanie et al., 2007* | WT (*Sp* D39 Δ*cps*) | Wild-type *S. pneumoniae* D39 Δ*cps* |
| Strain, strain background (*Sp*) | *Sp* (D39 Δ*cps*) | *Fenton et al., 2016* | AKF_Spn001 | Δ*bgaA::kan* |
| Strain, strain background (*Sp*) | *Sp* (D39 Δ*cps*) | *Fenton et al., 2016* | AKF_Spn002 | Δ*bgaA::add9*(*spec*) |
| Strain, strain background (*Sp*) | *Sp* (D39 Δ*cps*) | *Fenton et al., 2016* | AKF_Spn003 | Δ*bgaA::tetM*(*tet*) |
| Strain, strain background (*Sp*) | *Sp* (D39 Δ*cps*) | *Fenton et al., 2016* | AKF_Spn004 | Δ*bgaA::cat* |
| Strain, strain background (*Sp*) | *Sp* (D39 Δ*cps*) | *Fenton et al., 2016* | AKF_Spn005 | Δ*bgaA::erm* |
| Strain, strain background (*Sp*) | *Sp* (D39 Δ*cps*) | *Fenton et al., 2016* | AKF_Spn351 | Δ*lytA::cat* |
| Strain, strain background (*Sp*) | *Sp* (D39 Δ*cps*) | *Flores-Kim et al., 2019* | AKF_Spn704 | Δ*lytA::erm* |
| Strain, strain background (*Sp*) | *Sp* (D39 Δ*cps*) | *Flores-Kim et al., 2019* | JFK_SPN001 | Δ*lytA::erm* Δ*tacL::cat* |
| Strain, strain background (*Sp*) | *Sp* (D39 Δ*cps*) | *Flores-Kim et al., 2019* | JFK_SPN004 | *lytA*(H26A), *erm* |
| Strain, strain background (*Sp*) | *Sp* (D39 Δ*cps*) | *Flores-Kim et al., 2019* | JFK_SPN006 | *tacL*-FLAG, *spec* |
| Strain, strain background (*Sp*) | *Sp* (D39 Δ*cps*) | *Flores-Kim et al., 2019* | JFK_SPN008 | *tacL*-FLAG, *spec* Δ*lytA::erm* |
| Strain, strain background (*Sp*) | *Sp* (D39 Δ*cps*) | *Flores-Kim et al., 2019* | JFK_SPN013 | Δ*bgaA::*($P_{Zn}$-*tacL*, *tet*) Δ*tacL::cat* Δ*lytA::erm* |
| Strain, strain background (*Sp*) | *Sp* (D39 Δ*cps*) | This study | JFK_SPN014 | Δ*bgaA::*($P_{Zn}$-*lytA*, *tet*) Δ*lytA::erm* |
| Strain, strain background (*Sp*) | *Sp* (D39 Δ*cps*) | This study | JFK_SPN015 | Δ*bgaA::*($P_{Zn}$-*lytA*, *tet*) Δ*lytA::erm* Δ*whyD::spec* |

*Continued on next page*

*Continued*

| Reagent type (species) or resource | Designation | Source or reference | Identifiers | Additional information |
|---|---|---|---|---|
| Strain, strain background (*Sp*) | *Sp (D39 Δcps)* | This study | JFK_SPN016 | *ΔlytA::erm ΔwhyD::spec* |
| Strain, strain background (*Sp*) | *Sp (D39 Δcps)* | This study | JFK_SPN017 | *ΔbgaA::(P$_{Zn}$-whyD, tet)* |
| Strain, strain background (*Sp*) | *Sp (D39 Δcps)* | This study | JFK_SPN018 | *ΔbgaA::(P$_{Zn}$-whyD, tet) ΔwhyD::spec* |
| Strain, strain background (*Sp*) | *Sp (D39 Δcps)* | This study | JFK_SPN019 | *ΔbgaA::(P$_{Zn}$-whyD, tet) ΔwhyD::spec ΔlytA::erm* |
| Strain, strain background (*Sp*) | *Sp (D39 Δcps)* | This study | JFK_SPN020 | *ΔbgaA::(P$_{Zn}$-gfp-whyD, tet)* |
| Strain, strain background (*Sp*) | *Sp (D39 Δcps)* | This study | JFK_SPN021 | *ΔbgaA::(P$_{Zn}$-gfp-whyD, tet) ΔwhyD::spec* |
| Strain, strain background (*Sp*) | *Sp (D39 Δcps)* | This study | JFK_SPN022 | *ΔbgaA::(P$_{Zn}$-gfp-whyD, tet) ΔlytA::erm ΔwhyD::spec* |
| Strain, strain background (*Sp*) | *Sp (D39 Δcps)* | This study | JFK_SPN023 | *ΔbgaA::(P$_{Zn}$-gfp-whyD, tet) ΔwhyD::spec* |
| Strain, strain background (*Sp*) | *Sp (D39 Δcps)* | This study | JFK_SPN024 | *ΔbgaA::(P$_{Zn}$-whyD, tet) spd1526-1527::PF6-optRBS-gfp, kan* |
| Strain, strain background (*Sp*) | *Sp (D39 Δcps)* | This study | JFK_SPN025 | *ΔlytA::erm ΔlytC::tet ΔlytB::spec* |
| Strain, strain background (*Sp*) | *Sp (D39 Δcps)* | This study | JFK_SPN026 | *spd1526-1527::PF6-optRBS-gfp, kan* |
| Strain, strain background (*Sp*) | *Sp (D39 Δcps)* | This study | JFK_SPN027 | *ΔlytA::erm ΔlytC::tet ΔlytB::spec spd1526-1527::PF6-optRBS-gfp, kan* |
| Strain (*Bacillus subtilis*) | PY79 | *Youngman et al., 1983* | PY79 – wild-type strain | PY79 – wild-type strain |
| Strain (*Escherichia coli*) | DH5a | Gibco BRL | DH5a | *F-hsdR17 Δ(argF-lacZ)U169 phoA glnV44 Φ80dlacZ Δ M15 gyrA96 recA1 endA1 thi-1 supE44 deoR* |
| Strain (*E. coli*) | BL21(DE3) | New England Biolabs | BL21(DE3) | *E. coli* B F⁻ *ompT gal dcm lon hsdS$_B$(r$_B$⁻m$_B$⁻) [malB⁺]$_{K-12}$(λ ˢ) ΔfhuA* |
| Recombinant DNA reagent | plasmid | *Fenton et al., 2016* | pLEM023 | *bgaA'::P$_{zn}$::MCS::tetM::bgaA' bla* |
| Recombinant DNA reagent | plasmid | Novagen | pET24 | P$_{T7}$, *lacI$^q$*; vector used for protein expression |
| Recombinant DNA reagent | plasmid | This study | pER111 | *spd1526-1527::kan* |
| Recombinant DNA reagent | plasmid | *Flores-Kim et al., 2019* | pJFK001 | *tacL in pLEM023* |
| Recombinant DNA reagent | plasmid | *Flores-Kim et al., 2019* | pJFK002 | *lytA in pet24* |
| Recombinant DNA reagent | plasmid | This study | pJFK003 | *whyD in pLEM023* |
| Recombinant DNA reagent | plasmid | This study | pJFK004 | *lytA\* (H26A) in pET24* |
| Recombinant DNA reagent | plasmid | This study | pGD147 | *gfp-whyD in pLEM023* |
| Recombinant DNA reagent | plasmid | *Uehara et al., 2010* | pTD68 | P$_{T7}$, *lacI$^q$*; vector used for protein expression and for making 6xhis-SUMO fusions |
| Recombinant DNA reagent | plasmid | This study | pJFK005 | *whyD$^{CT}$ in pTD68* |
| Recombinant DNA reagent | plasmid | This study | pGD160 | *spd1526-1527::PF6-optRBS-gfp (kan)* |
| Recombinant DNA reagent | plasmid | *Wang et al., 2014* | pBB283 | *yhdG::kan* |
| Recombinant DNA reagent | plasmid | *Liu et al., 2017* | pPEPY-Pf6-*lacI* | Pf6-lacI expression vector used for lacI integration and expression from Sp chromosome |
| Antibody | WhyD (rabbit polyclonal) | This study | anti-WhyD | WB (1:10000) |
| Antibody | FLAG (TacL-FLAG; rabbit polyclonal) | Sigma | *RRID:AB_796202* | WB (1:5000) |
| Antibody | LTA/anti-Phosphocholine TEPC-15 (mouse monoclonal) | Sigma | *RRID:AB_1163630* | WB (1:1000) |

## Strains, plasmids, and growth conditions

All *Sp* strains were derived from the unencapsulated strain (D39 *Δcps*) (*Lanie et al., 2007*). Cells were grown in Todd Hewitt (Beckton Dickinson) medium supplemented with 0.5% yeast extract (THY) at 37 °C in an atmosphere containing 5% $CO_2$ or on pre-poured tryptic soy agar 5% sheep blood plates (TSAII 5% sheep blood, Beckton Dickinson) with a 5 ml overlay of 1% nutrient broth (NB) agar containing the required additives. When required, TSA agar plates containing 5% defibrinated sheep blood (Northeast laboratory) were used. *E. coli* strains were grown on Luria-Bertani (LB) broth or LB agar. WT *Bacillus subtilis* strain PY79 (*Youngman et al., 1983*) was grown in LB broth or LB agar as described previously (*Fenton et al., 2016*; *Fenton et al., 2018*; *Flores-Kim et al., 2019*). For both *S. pneumoniae* and *E. coli,* antibiotics were used as previously described. A list of strains and plasmids, and oligonucleotides used in this study can be found in the Key Resources Table and in *Supplementary file 2*, respectively.

## Transformation of *S. pneumoniae*

Transformations were performed as described (*Fenton et al., 2016*; *Fenton et al., 2018*; *Flores-Kim et al., 2019*). Briefly, cells in early exponential phase were back-diluted to an optical density at 600 nm ($OD_{600}$) of 0.03 using an Ultrospec 2100 spectrophotometer (Biochrom) and competence was induced with 500 pg/ml competence-stimulating peptide 1 (CSP-1), 0.2% BSA, and 1 mM $CaCl_2$. Cells were transformed with 100 ng chromosomal or plasmid DNA and selected on TSAII overlay plates containing the appropriate additives.

## Growth curves

To monitor growth kinetics and autolysis, *Sp* cells in early exponential phase were diluted to an $OD_{600}$ of 0.025 and grown to mid-exponential phase in THY medium containing the appropriate additives at 37°C in an atmosphere containing 5% $CO_2$. Cells were diluted to $OD_{600}$ of 0.025 in THY with the indicated additives and growth was monitored by $OD_{600}$ every 30 min. The figures that report growth curves are representative of experiments that were performed on at least two independent samples.

## Library generation and transposon insertion sequencing (Tn-seq)

Tn-seq was performed as described previously (*Flores-Kim et al., 2019*). Two independently generated libraries in WT and Δ*lytA* were used and reanalyzed in this study. Briefly, genomic DNA mutagenized with the Magellan6 transposon was transformed into competent *Sp*. Approximately 302,000 (wt) and 305,000 (Δ*lytA*) transformants were recovered for each library. Genomic DNA was then isolated and digested with MmeI, followed by adapter ligation. Transposon-chromosome junctions were PCR-amplified and sequenced on the Illumina HiSEq 2500 platform using TruSeq Small RNA reagents (Tufts University Core Facility Genomics). Reads were de-multiplexed, trimmed, and transposon insertion sites mapped onto the D39 genome. After normalization, a Mann-Whitney U-test was used to identify genomic regions with significant differences in transposon insertions. Transposon insertion profiles were visualized using the Artemis genome browser (v10.2).

## Isolation and analysis of pneumococcal LTAs

*Sp* strains were grown in THY medium with required additives at 37 °C in 5% $CO_2$ to the indicated growth phase and normalized to an $OD_{600}$ of 0.5. About 20 ml of the normalized culture were collected by centrifugation at 5000 × g for 5 min and the cell pellet was washed twice with 2 ml SMM (0.5M sucrose, 20mM maleic acid pH 6.5, 20 $MgCl_2$) and then re-suspended in 2 ml SMM. Protoplasts were generated by addition of lysozyme (1 mg/ml final concentration) and 100 units mutanolysin (sigma) and incubation at 37°C for 30 min. Protoplast formation was monitored by light microscopy. Protoplasts were pelleted by centrifugation at 5000 × g for 5 min and resuspended in 2 ml cold hypotonic buffer (20 mM HEPES ($Na^+$) pH 8.0, 100 mM NaCl, 1 mM dithiothreitol [DTT]), 1 mM $MgCl_2$, 1 mM $CaCl_2$, 2× complete protease inhibitors (Roche), 6 µg/ml RNAse A, 6 µg/ml DNAse. Unbroken protoplasts were removed by centrifugation at 20,000 × g for 10 min, and the lysate was then subjected to ultracentrifugation at 100,000 × g for 1 hr at 4°C. Membrane pellets were resuspended in 1 ml Tris-tricine sample buffer (200 mM Tris-HCl pH 6.8, 40% glycerol, 2% SDS, 0.04% Coomassie Blue G-250), boiled for 10 min, and analyzed by Tris-tricine PAGE followed by immunoblotting using monoclonal

antibody clone TEPC-15 (Sigma). The immunoblots in figures analyzing LTA levels are representative of experiments that were performed on at least two independently collected samples.

## Isolation and analysis of pneumococcal WTAs

*Sp* strains were grown and harvested as above. The pellets were resuspended in 2 ml of buffer 1 (50 mM 2-[*N*-morpholino ethanesulfonic acid [MES]] pH 6.5) and centrifuged at 7,000 × g for 5 min. The resulting pellets were resuspended in 2 ml buffer 2 (50 mM MES pH 6.5, 4% [w/v] SDS) and incubated in boiling water for 1 hr. The sample was then centrifuged at 7000 × g for 5 min and the pellet was washed with 2 ml buffer 2. The sample was transferred into a clean microfuge tube and centrifuged at 16,000 × g for 5 min. The pellet was then washed with 2 ml buffer 2, followed by successive washes with 2 ml buffer 3 (50 mM MES pH 6.5, 2% [w/v] NaCl) and 2 ml buffer 1. The samples were then centrifuged at 16,000 × g for 5 min, resuspended in 2 ml of buffer 4 (20 mM Tris-HCl pH 8.0, 0.5% [w/v] SDS) supplemented with 2 μl proteinase K (20 mg/ml), and incubated at 50°C for 4 hr with shaking (1000 rpm). The pellet was then collected by centrifugation and washed with 2 ml buffer 3 followed by 3 washes with distilled water. The pellet was collected by centrifugation and subjected to alkaline hydrolysis in 0.5 ml of 0.1 N NaOH at 25 °C for 16 hr with shaking (1000 rpm). The samples were then pelleted by centrifugation and the supernatants containing the extractable WTA were collected and resuspended in 0.5 ml native sample buffer (62.5 mM Tris-HCl pH 6.8, 40% glycerol, 0.01% bromophenol blue). Samples were analyzed by native PAGE followed by alcian blue-silver staining. The gels in figures analyzing WTA levels are representative of experiments that were performed on at least two independently collected samples.

## Purification of LytA (rLytA) and LytA* (rLytA) and labeling with Alexa Fluor594

rLytA and rLytA* were overexpressed in *E. coli* BL21(DE3) Δ*fhu*A2 (New England Biolabs) containing the pET21amp-*lytA* or pET21amp-*lytA*\* expression vectors. Cells were grown in LB supplemented with 100 μg/ml ampicillin at 37°C and expression was induced at an $OD_{600}$ of 0.5 with 1 mM IPTG for 2 hr at 37 °C. Cells were collected by centrifugation and stored overnight at –20 °C. The cell pellets were resuspended in lysis buffer (20 mM Tris-HCl pH 7.5, 500 mM NaCl, 200 μg/ml DNase, and 2×complete protease inhibitors [Roche]) and lysed by two passages through a cell disruptor (Constant systems Ltd.) at 25,000 psi. Unbroken cells were discarded by centrifugation. The supernatant was then passed over a DEAE cellulose column (sigma). After washing with 20 column volumes of wash buffer (20 mM $NaPO_4$ pH 7, 1.5 M NaCl), LytA was eluted with 2 column volumes of wash buffer supplemented with 140 mM choline chloride. Protein-containing fractions were pooled and dialyzed against 20 mM $NaPO_4$ pH 7.5, 150 mM NaCl, 10% glycerol, and 5 mM choline chloride. Purified rLytA and rLytA* were labeled with the Alexa Fluor594 protein labeling kit according to manufacturer instructions (Thermo Fisher Scientific).

## Purification of WhyD$^{CT}$ and antibody production

The C-terminal domain of WhyD (WhyD$^{CT}$) was expressed in *E. coli* BL21(DE3) Δ*fhuA* using the $P_{T7}$-His$_6$-SUMO-*whyD$^{CT}$* expression vector (pTD68-*whyD*). Cells were grown in LB supplemented with 100 μg/ml ampicillin at 37°C to an $OD_{600}$ of 0.5. Cultures were allowed to equilibrate at room temperature for 30 min and then transferred to 30°C. *his$_6$-sumo-whyD$^{CT}$* expression was induced with 0.5 mM IPTG for 3 hr. Cells were collected by centrifugation, resuspended in 50 ml Buffer A (100 mM Tris-HCl pH 8.0, 500 mM NaCl, 20 mM Imidazole, and 2× complete protease inhibitor tablets [Roche]), and stored at −80°C. The cell suspension was thawed on ice and lysed by two passes through a cell disruptor at 25,000 psi. The lysate was clarified by ultracentrifugation at 35 Krpm for 30 min at 4°C. The supernatant was added to 1 ml Ni$^{2+}$-NTA resin (Qiagen) and incubated for 1 hr at 4°C. The suspension was loaded into a 10 ml column (BioRad), washed twice with 4 ml Buffer A, and eluted with 2.5 ml Buffer B (100 mM Tris-HCl pH 8.0, 500 mM NaCl, 300 mM Imidazole). 10 μL of purified His$_6$-Ulp1 protease (1.25 mg/ml) was added to the eluate, and the mixture was dialyzed into 100 mM Tris-HCl pH 8, 100 mM NaCl, 10% glycerol overnight at 4°C. The next morning 10 μl more His$_6$-Ulp1 was added to the dialysate and incubated for 1 hr at 30°C. The dialysate was mixed with 1 ml of Ni$^{2+}$-NTA resin for 1 hr at 4°C and then loaded onto a column and the WhyD$^{CT}$-containing flow-through was collected, dialyzed into 100 mM Tris-HCl pH 8, 100 mM NaCl, 1 mM CaCl$_2$, 10% glycerol overnight at 4°C and

stored at −80°C. The purified protein was used for in vitro assays and to generate rabbit polyclonal antibodies (Covance).

## Immunoblot analysis

*Sp* cultures were normalized to an $OD_{600}$ of 0.5 and harvested. Cell extracts were prepared by resuspension of cell pellets in lysis buffer (20 mM Tris pH 7.5, 10 mM EDTA, 1 mg/ml lysozyme, 100 units mutanolysin [Sigma]) 10 µg/ml DNase I, 100 µg/ml RNase A, and 2× complete protease inhibitors (Roche Applied Sciences) and incubation at 37°C for 10 min. SDS sample buffer (100 µl, 0.25 M Tris pH 6.8, 4% SDS, 20% glycerol, 10 mM EDTA) containing 10% 2-mercaptoethanol was added to each preparation. Proteins were resolved by SDS- or Tris-tricine-PAGE and transferred to nitrocellulose membranes by semidry or wet-transfer immunoblotting as indicated. The membranes were probed with anti-WhyD diluted 1:10,000, anti-FLAG (Sigma) diluted 1:1, 5000, or the monoclonal antibody TEPC-15 (Sigma) diluted 1:1000. Primary antibodies were detected with goat anti-rabbit IgG or goat anti-mouse IgG (Bio-Rad) horseradish peroxidase conjugate used at a 1:10,000 dilution. Secondary antibodies were detected by enhanced chemiluminescence on an Azure Biosystems C600 gel-Doc and western blot imaging system.

## In vitro WTA and LTA release assays using WhyD$^{CT}$

WhyD$^{CT}$ activity was assayed using purified sacculi (from Δ*lytA*Δ*whyD* cells to obtain larger quantities of WTAs attached to sacculi) prepared as described above omitting the alkaline hydrolysis step. The release assays were conducted with 0.1 mg sacculi and 10 µg/ml WhyD$^{CT}$, 10 µg/ml WhyD$^{CT}$ +1 mM EDTA, or no WhyD$^{CT}$ in 1 ml reaction buffer (0.1 M Tris-HCl pH 8, 1 mm $CaCl_2$) incubated at room temperature with gentle shaking. Released WTAs were collected by centrifugation. To recover WTAs that were not released, the sacculi pellets were then treated with 0.1 M NaOH overnight at room temperature with gentle shaking. Alkaline-released WTAs were collected by centrifugation and analyzed alongside the WhyD-released WTAs by SDS-PAGE followed by alcian blue-silver staining.

LTA assays were performed in reaction buffer with 0.1 mg homogenized membrane extracts (from Δ*lytA* Δ*whyD* cells) prepared as described above. 0.1 mg of the homogenized membranes were incubated with 10 µg/ml WhyD$^{CT}$, 10 µg/ml WhyD$^{CT}$ +1 mM EDTA, or no WhyD$^{CT}$ in 1 ml reaction buffer (0.1 M Tris-HCl pH 8, 1 mm $CaCl_2$), and incubated at room temperature with gentle shaking. After incubation, the reactions were quenched with 1 mM EDTA. Released and membrane-associated LTAs were then analyzed by 16% Tris-tricine SDS-PAGE and probed with a monoclonal antibody specific for phosphocholine. The data presented in the figures are representative of experiments that were performed on at least two independently collected samples.

## In vivo WTA labeling experiments using LytA*⁻Alexa in *Sp* and *B. subtilis*

Strains were grown to mid-exponential phase and labeled for 5 min with HADA. The cells were then washed with fresh medium and labeled for 5 min with sBADA. The equivalent of 1 ml of cells at $OD_{600}$ of 0.5 was washed with fresh medium containing 1% choline and then was incubated with 1 µg/ml rLytA*-Alexa for 30 s with gentle shaking. Cells were washed twice with 1 × PBS and analyzed by fluorescence microscopy. The micrographs in the figures are representative of experiments that were performed on at least three independently collected samples.

## *Sp pneumoniae* sacculi preparation and labeling

Cells were grown to mid-exponential phase, labeled with sBADA for 5 min, and sacculi with or without WTAs were prepared as described above. Sacculi from the equivalent of 1 ml of cells at $OD_{600}$ of 0.5 were labeled with 1 µg/ml rLytA*-Alexa as described above the samples were imaged by fluorescence microscopy. The micrographs in the figures are representative of experiments that were performed on at least three independently collected samples.

## Fluorescence microscopy

Cells were harvested and concentrated by centrifugation at 6800 × g for 1.5 min, re-suspended in 1/10th volume growth medium, and then immobilized on 2% (wt/vol) agarose pads containing 1× PBS. Fluorescence microscopy was performed on a Nikon Ti inverted microscope equipped with a

Plan Apo 100×/1.4 Oil Ph3 DM phase contrast objective, an Andor Zyla 4.2 Plus sCMOS camera, and Lumencore SpectraX LED Illumination. Images were acquired using Nikon Elements 4.3 acquisition software. HADA was visualized using a Chroma ET filter cube for DAPI (49000); sBADA and GFP were visualized using a Chroma ET filter cube for GFP (49002); LytA*-AlexaFluor594 was visualized using a Chroma ET filter cube for mCherry (49008). Image processing was performed using Metamorph software (version 7.7.0.0) and quantitative image analysis was performed using Oufti (*Paintdakhi et al., 2016*). Cytoplasmic fluorescent signal was used to facilitate cell segmentation in Oufti, and PG incorporation as monitored by FDAAs was used to monitor cell cycle progression.

## Structured illumination microscopy (SIM)

Acquisitions were performed on an Elyra 7 system with SIM$^2$, with dual PCO Edge 4.2 sCMOS cameras, on an inverted microscope via a motorized Duolink camera adapter (Carl Zeiss Microscopy, Jena Germany, and PCO, Kelheim, Germany). Channels were set up as two line-switched imaging tracks with 405 and 561 acquired simultaneously in one track, and 488 on a separate line-switched track. Samples were imaged with a Plan Apochromat 63×/1.4 oil objective with immersion medium Immersol 518 F (30°C) with a 1.6× optovar in the light-path providing 60 nm acquisition pixel spacing. Emission windows were set via a combination of a long-pass 560 nm dichroic in the light-path, and dual-pass emission filters (420–480+495-550 nm; and 570–620+LP 655 nm) for the two cameras. A single grid spacing for both tracks was automatically selected by the Zen 3.0 software based on the wavelength and objective combination. Nine SIM phases were acquired for each planar step per channel, while Z-stacks were acquired with a spacing of 270 nm in the 'Leap Mode' over a range of 4–5 μm (https://www.zeiss.com/microscopy/us/products/super-resolution/elyra-7.html). Laser powers were set up to achieve ~3000 gray values in the 16-bit raw image per channel. SIM$^2$ processing was performed based on theoretical PSF's using the built-in unified SIM$^2$ processing function in Zen, with 25 iterations of the Constrained Iterative method at a regularization weight of 0.015, with processing pixel-sampling scale-up resulting in 16 nm pixel sizes in x–y, and 90 nm step-size in z.

## Quantification and statistical analyses

Cell sizes were calculated using meshes generated by Oufti (*Paintdakhi et al., 2016*) and the Matlab script getCellDimensions. Cell fluorescence, normalized to cell area, was calculated using meshes generated by Oufti the Matlab script getSignal1. Demographs were generated using Oufti's built-in demograph feature (*Paintdakhi et al., 2016*). Fluorescence intensity profiles along the cell length were generated using meshes generated by Oufti and the Matlab script signal1_alonglength. For statistical comparisons of cell size between strains, unpaired t-tests with Welch's correction were performed. Experiments were performed 3 independent times (N=3). For all tests, at least 300 cells were used for image analysis. Scripts, output data, and calculated p-values can be found in the source data files associated with each figure.

## Strain construction

### *Sp* deletion strains

All *Sp* deletion strains were generated using PCR fragments as described previously and are listed in the Key Resources table. Briefly, two products representing the regions (~1 kb each) flanking the target gene were amplified, and an antibiotic resistance cassette ligated between them using Gibson assembly. Assembled PCR products were transformed directly into *Sp* as described above. In all cases, deletion primers were given the name: 'gene name'_5FLANK_F/R for 5′ regions and 'gene name'_3FLANK_F/R for 3′ regions. Antibiotic markers were amplified from *ΔbgaA::antibiotic cassette* (*bgaA* gene disrupted with an antibiotic cassette) strains using the AB_Marker_F/R primers. A full list of primer sequences can be found in the *Supplementary file 2*. Extracted gDNA from deletion strains was confirmed by PCR using the AntibioticMarker_R primer in conjunction with a primer binding ~ 200 bp 5′ of the disrupted gene; these primers were given the name: 'gene name'_Seq_F. Confirmed gDNAs of single gene deletions were used for the construction of multiple knockout strains. For strains containing multiple deletions and construct integrations, transformants were verified by re-streaking on media containing the relevant antibiotics. When needed, each construct was confirmed by diagnostic PCR and/or sequencing.

### P$_{zn}$-lytA

The *lytA* ORF, with its native RBS, was amplified using primers *lytA_F_nativeRBS_XhoI* and *lytA_R_ BamHI*. The primers introduced XhoI and BamHI sites used for cloning into pLEM023 (**Fenton et al., 2016**) cut with the same enzymes, resulting in plasmid pJFK004. The plasmid was sequenced and used to transform strain D39 *Δcps ΔbgaA::kan lytA::erm*. Integration into the *bga* locus was confirmed by antibiotic marker replacement and PCR using the *bgaA_FLANK_F* and *bgaA*_FLANK_R primers. gDNA from the resulting strain was prepared and then used to transform the appropriate *Sp* strains.

### P$_{zn}$-whyD

The *whyD* ORF, with its native RBS, was amplified using primers *whyD_F_optRBS_XhoI* and *whyD_R_ BamHI*. The primers introduced XhoI and BamHI sites used for cloning into pLEM023 cut with the same enzymes, resulting in plasmid pJFK003. The plasmid was sequenced and used to transform strain D39 *Δcps Δbga::kan*. Integration into the *bga* locus was confirmed by antibiotic marker replacement and PCR using the *bgaA_FLANK_F* and *bgaA*_FLANK_R primers. gDNA from the resulting strain was prepared and then used to transform the appropriate *Sp* strains.

### *P$_{zn}$-gfp-whyD* (pGD147)

The plasmid was generated in a 3-piece isothermal assembly reaction with (1) a PCR product containing *gfp* with an optimized RBS (oligonucleotide primers oGD267/268), (2) a PCR product containing *whyD* (oligonucleotide primers oGD369/270), and (3) pLEM023 digested with XhoI and BamHI. The resulting construct was sequence-confirmed and used to transform strain D39 *Δcps Δbga::kan*. Integration into the *bga* locus was confirmed by antibiotic marker replacement and PCR using the *bgaA_FLANK_F* and *bgaA_FLANK_R* primers. gDNA from the resulting strain was used to transform the appropriate *Sp* strains.

### *pET24-lytA*$^{H26A}$ (pET24-lytA*)

The *lytA* ORF was amplified using primers *lytA_F_purification_NdeI* and *lytA_R_purification_HindIII* from the chromosome of D39 *Δcps lytA-H26A* (JFK_Spn004). The primers introduced NdeI and HindIII sites used for cloning into peT24a cut with the same enzymes, resulting in plasmid pJFK004. The plasmid was confirmed by sequencing.

### pTD68-whyD$^{CT}$

The *whyD*$^{CT}$ ORF was amplified using primers *whyD_CTERM_BamHI_F_pTD68* and *whyD_CTERM_ XhoI_R_pTD68* from the chromosome of D39 *Δcps*, and cloned into pTD68 (**Uehara et al., 2010**) cut with the same enzymes to generate pJFK005. The plasmid was confirmed by sequencing.

### pER111

The plasmid was generated using a 2-piece ligation reaction containing spd1526'–1527' from pER87 (unpublished) and the kanamycin cassette digested from pBB283 (**Wang et al., 2014**) using BglII/SalI. The resulting construct was sequence-confirmed.

pER111-Pf6-gfp pGD160 was generated using a 3-piece isothermal assembly reaction containing (1) a PCR product containing the PF6 promoter and an optimized RBS (oligonucleotide primers oGD391/392 from pPEPY-PF6-lacI **Liu et al., 2017**), (2) a PCR product containing gfp (oligonucleotide primers oGD67/193), and (3) pER111 digested with XhoI and BamHI. The resulting construct was sequence-confirmed. The plasmid was sequenced and used to transform strain D39 *Δcps* Integration into spd1526-1527 locus was confirmed by PCR using the primers oSp95 and oSp98. gDNA from the resulting strain was prepared and then used to transform the appropriate Sp strains.

## Acknowledgements

We thank all members of the Bernhardt and Rudner laboratories past and present for support and helpful comments. We thank Hoong Chuin Lim, Joel Sher, and Andrea Vettiger for helpful insights into data analysis, Paula Montero Llopis and her team at the Microscopy Resources on the North Quad (MicRoN) core facility at Harvard Medical School for microscopy services, Eammon Riley for pER111, Simosenkosi Nkomboni for technical assistance, and Erkin Kuru for advice on FDAA labeling.

We acknowledge Tanay Desai and Geoff Guimaraes of Carl Zeiss Microscopy for sample runs on the Elyra 7 with $SIM^2$ and Radhika Khetani and the HMS Chan bioinformatics core for technical assistance. JFK is a jointly mentored postdoctoral fellow bridging work in both the Bernhardt and Rudner labs and was funded in part by the Ruth L Kirschstein Postdoctoral Individual National Research Service Award from the National Institutes of Health (NIH-NIAID, F32AI36431). This work was also supported by the Howard Hughes Medical Institute (TGB) and the National Institutes of Health grants R01AI083365 (TGB), R01GM127399 and R01GM086466 (DZR), and R01AI139083 (TGB and DZR). GSD was supported, in part, by the T32 Bacteriology PhD Training Program (T32AI132120).

## Additional information

### Funding

| Funder | Grant reference number | Author |
| --- | --- | --- |
| National Institute of Allergy and Infectious Diseases | F32AI36431 | Josué Flores-Kim |
| National Institute of Allergy and Infectious Diseases | R01AI083365 | Thomas G Bernhardt |
| Howard Hughes Medical Institute | | Thomas G Bernhardt |
| National Institute of General Medical Sciences | R01GM127399 | David Z Rudner |
| National Institute of General Medical Sciences | R01GM086466 | David Z Rudner |
| National Institute of Allergy and Infectious Diseases | R01AI139083 | Thomas G Bernhardt David Z Rudner |
| National Institute of Allergy and Infectious Diseases | T32AI132120 | Genevieve S Dobihal |

The funders had no role in study design, data collection and interpretation, or the decision to submit the work for publication.

### Author contributions

Josué Flores-Kim, Genevieve S Dobihal, Conceptualization, Formal analysis, Investigation, Methodology, Validation, Writing – original draft; Thomas G Bernhardt, David Z Rudner, Conceptualization, Funding acquisition, Project administration, Supervision, Writing – original draft

### Author ORCIDs

Josué Flores-Kim ![ORCID] http://orcid.org/0000-0001-8282-6647
Genevieve S Dobihal ![ORCID] http://orcid.org/0000-0001-7589-1133
Thomas G Bernhardt ![ORCID] http://orcid.org/0000-0003-3566-7756
David Z Rudner ![ORCID] http://orcid.org/0000-0002-0236-7143

### Decision letter and Author response

Decision letter https://doi.org/10.7554/eLife.76392.sa1
Author response https://doi.org/10.7554/eLife.76392.sa2

## Additional files

### Supplementary files

• Supplementary file 1. Test of suppression of *whyD* essentiality by deletion of genes encoding choline binding cell wall hydrolases. Transformations with indicated amplicons were performed as described in Materials and methods. Each transformation experiment was performed 3 times independently with similar results and the average number of colonies from the 3 experiments are shown. * Direct transformation of a *ΔwhyD::spec* amplicon into the indicated stains resulted in tiny colonies that were unable to grow in liquid media after 10 hr.

- Supplementary file 2. Oligonucleotides used in this study.
- Transparent reporting form

## Data availability

All data generated or analysed during this study are included in the manuscript and supporting file; source data files have been provided for all figures.

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
