## [Editor Report]

The mechanisms by which cell wall hydrolases are controlled to prevent their lethal misactivation are not well understood. This study reports the identification and characterization of an enzyme, named WhyD, that specifically hydrolyzes wall teichoic acids (TAs) in Streptococcus pneumoniae. Importantly, WhyD activity depletes TA content at active PG synthesis sites. Because the major autolysin LytA binds to TAs, this mechanism prevents its action at these sites.

---

## [Decision Letter]

**Decision letter after peer review:**

Thank you for submitting your article "WhyD tailors surface polymers to prevent bacteriolysis and direct cell elongation in *Streptococcus pneumoniae*" for consideration by *eLife*. We apologize for the delay it has taken in making a decision. Your article has now been reviewed by 3 peer reviewers, and the evaluation has been overseen by a Reviewing Editor and Gisela Storz as the Senior Editor. The following individual involved in the review of your submission has agreed to reveal their identity: Alexandre W Bisson Filho (Reviewer #2).

Essential revisions:

1) The proposed model appropriately explains the regulation of autolysis in exponentially grown cells. However, cells still undergo autolysis in late stationary phase which does not appear to be WhyD-dependent. It would be good to make sure that the narrative clearly acknowledges this puzzling observation; the title seems a bit misleading and the discussion should clearly indicate this current gap in our knowledge.

2) The authors model suggest that the activity of WhyD would reduce the activity of any cell wall hydrolase that also bind choline. In other words, can lethality of whyD mutations be also reverted in genetic mutants other than lytA, for example cbpD? Does whyD deletion lead to second site suppressors that affect other hydrolases?

3) The microscopy data needs better quantitation and visualization. For example, it's not clear whether there were replicates, the sample size (informing that at least 300 cells were used is not enough information to inform on sample size effects). Sub-panels where no signal is apparently detected (e.g. Figure 7 and supplements) should be clarified and the background should be displayed.

*Reviewer #1 (Recommendations for the authors):*

The proposed model to explain the control of autolysis in exponentially growing cells (and during early stationary phase) is convincing and the data presented nicely support it. As (briefly) stated in the text, cells still undergo autolysis in late stationary phase despite the production of WhyD, indicating that (an)other mechanism(s) exist(s) to control peptidoglycan hydrolysis. It would be good to make sure that the narrative clearly acknowledges this puzzling observation; the title seems a bit misleading and the discussion should clearly indicate this current gap in our knowledge.

The data show that despite the production of WhyD during autolysis, WTAs are detected, suggesting that the enzyme is inactive (or present in insufficient amounts). Looking at the Western Blots in figure 2 —figure supplement 1, it is difficult to know what panel A is showing (no molecular weights are shown). What does the cropped image show? Is this the full length protein? (why are the 2 bands recognized by the anti-WhyD serum in Figure 6B not seen in Figure 2 Supp 1?).

Several mechanisms could account for the lack of WhyD activity, including a proteolytic cleavage, releasing the catalytic domain away from nascent peptidoglycan or the lack of a co-factor. I certainly agree that WhyD is present throughout growth, but it is possible that the amount of protein decreases in autolytic phase. The authors should be more cautious and/or quantify the relative amount of WhyD throughout growth (using densitometry). "WhyD levels remain constant throughout growth and under autolytic conditions" is a bit assertive given the experimental evidence shown.

*Reviewer #2 (Recommendations for the authors):*

I really like the paper and I think there should not be much work to be done to get it ready for publication. I have my comments and suggestion in order as it shows in the manuscript:

Line 7: "Penicillin and related antibiotics disrupt cell wall synthesis in bacteria and induce lysis by misactivating cell wall hydrolases called autolysins"

This sentence should be re-written. As it is, it misleads non-experts to think that Penicillin stops cell wall synthesis BY directly activating hydrolases, resulting in lysis. That's not the broad mechanism of Penicillin. The central problem with the sentence might be "misactivating". I understand that the conclusion of the paper solidifies the idea that, in the absence of proper cell wall synthesis, hydrolase misactivation results in lysis and this is important in the context of some antibiotics. But again, you should be careful not to confuse the general public target of this journal. This also re-appears in line 97.

Line 51: "We recently discovered that penicillin treatment of Sp causes a dramatic shift in surface polymer biogenesis in which cell wall-anchored teichoic acids (WTAs) increase in abundance at the expense of lipid-linked lipoteichoic acids".

You can simplify "lipid-linked lipoteichoic acids" by "lipoteichoic acids" as "lipo" already implies "lipid-linked". For consistency, you can ass "(LTA)" at the end to make the contrast with "WTA".

Line 54: "Because LytA binds to these polymers,…"

What polymer? PG? WTA? LTA? Please be specific.

Line 93: "They also play important roles in cleaving shared cell wall material connecting daughter cells during cytokinesis (Vollmer et al., 2008)".

I suggest adding this interesting and important paper as well: https://pubmed.ncbi.nlm.nih.gov/29458657/

Also, as far as I know, the most comprehensive work on the global deletion of all hydrolases in Bacillus and how the majority of them play a role in daughter cell separation. This same reference can be used in lines 107-110: https://www.biorxiv.org/content/10.1101/2021.02.18.431929v1

I also like the following: https://pubmed.ncbi.nlm.nih.gov/12399477/

Line 161: "Furthermore, LytA production in cells lacking WhyD during growth in liquid medium caused premature lysis in late exponential phase (Figure 1C)."

Why would this happen only in the late exponential/stationary phase? Or this is only detected in bulk growth at this stage but it's consistently happening throughout all phases?

Figure 1D: why the concentration of purified WhyD needs to be so high (1mg/ml)? If a titration was done, could be interesting to include the data and a sentence.

Line 204: "To test this possibility, we purified the C- terminal GlpQ domain of WhyD (WhyDCT) and monitored its ability to release WTAs from purified sacculi (Figure 3B)".

Make sure to clarify that the reason to purify the GlpQ sub-domain alone is because of the difficulty to purify it together with the transmembrane region.

Figures 4 and 5: I fail to understand what the GFP, HADA, and sBADA are doing there. And why sometimes do you use one or another probe? There is nothing about it in the text. Also, I strongly recommend including sample N in every plot. If possible, please change distributions to superplots where we can visualize the distributions of each biological replicate.

Figure 4: I noticed from the images that the major difference between different genetic backgrounds may come from cells in duplets/short chains compared to long chains. It would be interesting to point out if that's true (cell wall, WTA, and LTA composition may vary with cell-cell association). It should be easy to perform this analysis and correlate chain length and cell length/area/width. This could bring new insights on whether the effect seen in the previous experiments could come majorly from short chains (enriched in late exponential and stationary phases). Also, do short chains grow faster / slower / the same as long chains? Are they more sensitive to antibiotics?

Figure 5: if authors want to imply cells are smaller due to elongation defects, wouldn't be much easier to simply perform single-cell tracking from time-lapses instead of indirectly staining the cell wall?

Figure 6: Even though the rLytA-Alexa data is very compeling, I am not convinced GFP-WhyD is recruited to midcell. It could be simply that cells have double the membrane at midcell. Authors could make their point stronger by staining membranes and quantitating membrane:GFP-WhyD ratios to prove there is indeed accumulation of WhyD compared to the cell body.

Figure 7-Sup1-2-3: Please, make sure to show another panel with the background signal when LytA does not bind cells. It's very surprising that there is no signal at all. Even inespecific signal.

*Reviewer #3 (Recommendations for the authors):*

I have no major objection and only some suggestions that could improve the manuscript.

– Page 6, line 153 : WhyD is assumed to be essential but some deletions can be obtained in WT cells. Does suppressive mutations occurs only in lytA or other genes encoding hydrolases? Whole genome sequencing of some strains would help to strengthen authors hypothesis that WhyD modulates WTA localization and thus localization of hydrolases involved in cell elongation and final separation.

– To construct the ∆whyD∆lytA, I guess the author deleted whyD in the ∆lytA background? Was transformation efficiency similar to WT cells transformed with a control DNA? If not, it could be worth identifying the additional suppressive mutations to establish a genetic link with other hydrolases or factors promoting cell elongation.

– To be consistent with their previous binding, does whyD overexpression suppress at least the essentially of tacL (that requires a mutation in cozE)?

– Regarding the link between lytA and whyD. Did the author try to evaluate the essentiality of whyD in other hydrolases mutant that also bind choline? For instance, the hydrolase CbpD, that is expressed during competence in the early exponential phase, also possesses choline-binding domain. Are the same observations made in a ∆cbpD mutant?

– Page 8, lines 206-208: WhyD clearly releases WTA from PG sacculi, but a significant amount remains linked. Did the authors try to make longer incubation with more WhyD? it is possible that a fraction is resistant to WhyD hydrolysis? If yes, can the author comment on that? It could be due other modifications of PG (acetylation) or WTA (A-alanylation). It could make sense with the suppressive mutation detected upon deletion of whyD in WT cells (see my two first comments).

– Page 11 and Figure 6 : From the phase contrast image, it seems that cells chain. Does quantification confirm this visual impression? If yes, it would mean that the lower ectopic expression of GFP-WhyD (compared to whyD expression in WT cells (Fig6-Sup1B)) would also impact the activity of LytB? Does addition of purified recombinant LytB (as described in Rico-Lastress et al., Sci. Rep., 2015) abolish cell chaining? If yes, it would provide a direct link between WhyD and LytB.

– I would be great to analyze the dynamic of WhyD localization in the course of the cell cycle. Maybe the fluorescence signal is not compatible with time-lapse microscopy but sorting and plotting cells as a function of the different cell division stages may help to address this point and notably if WhyD localizes early or late at the division septum?

– Page 12, line 309: there is actually no rLytA* signal co-localizing with BADA in elongating cells in the Figure!

– Figure 7B: White head arrow could be colored in blue to be consistent with the color code of Figure 7A.

– Page 13, line 322: Figure 7D instead of Fig7C

– Figure 2-FigSup1 and others: What is the protein detected as the LC?

– Figure 3: an amino acid sequence alignment would be helpful to appreciate the homology with GlpQ phosphodiesterase domain.

---

## [Author Response]

Essential revisions:1) The proposed model appropriately explains the regulation of autolysis in exponentially grown cells. However, cells still undergo autolysis in late stationary phase which does not appear to be WhyD-dependent. It would be good to make sure that the narrative clearly acknowledges this puzzling observation; the title seems a bit misleading and the discussion should clearly indicate this current gap in our knowledge.

We agree with the reviewers' concerns. To address this, we have modified the Title, Abstract, and Discussion to reflect this current gap in our knowledge and to stress the importance of WhyD activity during exponential phase.

The Title has been modified "WhyD tailors surface polymers to prevent premature bacteriolysis and direct cell elongation in Streptococcus"

The Abstract was modified to emphasize the importance of WhyD during exponential phase as follows:

Line 57-58 now reads: In this report, we identify WhyD (SPD_0880) as a new factor that controls the level of WTAs in Sp cells to prevent LytA misactivation during exponential growth and premature lysis.

Line 60-62 now reads: Our results support a model in which the WTA tailoring activity of WhyD during exponential growth directs PG remodeling activity required for proper cell elongation in addition to preventing autolysis by LytA.

The relevant changes to the Discussion are:

Line 343-344 now reads: This pruning of WTAs prevents the excessive recruitment of the LytA PG hydrolase to the cell wall to avoid autolysis during exponential growth.

We have also added Lines 454-459 to clearly indicate that other mechanisms contribute to autolysis regulation during exponential and stationary phases, which require further investigation. These lines read:

“This process presumably also requires the release of LTAs from the membrane to prevent them from sequestering LytA away from the WTAs and PG (Flores-Kim et al., 2019). By contrast, WhyD appears to be stable under autolytic conditions (Figure 2 —figure supplement 1). Whether its activity must also be inhibited for autolytic induction and whether other mechanisms involved in controling autolysis exist remains unknown and requires further investigation.”

2) The authors model suggest that the activity of WhyD would reduce the activity of any cell wall hydrolase that also bind choline. In other words, can lethality of whyD mutations be also reverted in genetic mutants other than lytA, for example cbpD? Does whyD deletion lead to second site suppressors that affect other hydrolases?

We thank the reviewers for raising this interesting point. To test the contribution of other choline-binding hydrolases, we attempted to inactivate whyD in strains lacking one or several choline-binding hydrolases (single mutants: ∆lytB, ∆lytC, ∆cbpD; or the triple mutant lacking all three hydrolases). In all cases, transformations to introduce the ∆whyD mutant into these strains yielded >500-fold fewer transformants compared to transformations into a ∆lytA strain. Thus, additional suppressor mutations are likely needed to introduce the a DwhyD::spec allele into strains lacking choline-binding PG hydrolases other than LytA. We think this result shows that the lethality of WhyD inactivation principally arises through the misactivation of LytA. We now include these data as a supplementary table (Supplementary Table S1) and mention these results in the Discussion.

Lines 443-447 now read:

“We discovered WhyD based on its essential function in preventing LytA-induced autolysis of Sp cells (Figures 1, 2, Figure 2 —figure supplement 2, and supplementary Table 1). Inactivation of choline-binding hydrolase other than LytA were unable to suppress whyD essentiality (Supplementary Table 1), indicating that the misactivation of other hydrolases does not play a major role in the lethal phenotype of the ∆whyD mutant.”

3) The microscopy data needs better quantitation and visualization. For example, it's not clear whether there were replicates, the sample size (informing that at least 300 cells were used is not enough information to inform on sample size effects). Sub-panels where no signal is apparently detected (e.g. Figure 7 and supplements) should be clarified and the background should be displayed.

We thank the reviewers for requesting these data. We have included information about replicates (N number) in Materials and methods and have modified the figures to display more of the background in subpanels where no signal was detected.

Reviewer #1 (Recommendations for the authors):The proposed model to explain the control of autolysis in exponentially growing cells (and during early stationary phase) is convincing and the data presented nicely support it. As (briefly) stated in the text, cells still undergo autolysis in late stationary phase despite the production of WhyD, indicating that (an)other mechanism(s) exist(s) to control peptidoglycan hydrolysis. It would be good to make sure that the narrative clearly acknowledges this puzzling observation; the title seems a bit misleading and the discussion should clearly indicate this current gap in our knowledge.

We thank the reviewer for raising this point.

In our previous study we showed that in late stationary phase (or after antibiotic exposure), the LTA synthase is degraded. This has two consequences. (1) TA precursors that were previously used to synthesize LTAs become available for transfer to the PG resulting in an increase in WTA production. (2) Because LytA binds to the choline moieties on LTA and WTA, the reduction in LTAs enables LytA to bind WTA, causing autolysis. The data in this manuscript indicates that WhyD levels remain unchanged in late stationary phase. At present, we do not know whether WhyD activity is regulated and specifically reduced in late stationary phase or whether WhyD continues to remove WTAs but fails to keep pace with an increased level of production. We favor the former model as the latter seems wasteful.

We have elaborated on these points in the Abstract and Discussion. Please see our response to Essential Revision #1.

The data show that despite the production of WhyD during autolysis, WTAs are detected, suggesting that the enzyme is inactive (or present in insufficient amounts). Looking at the Western Blots in figure 2 —figure supplement 1, it is difficult to know what panel A is showing (no molecular weights are shown). What does the cropped image show? Is this the full length protein? (why are the 2 bands recognized by the anti-WhyD serum in Figure 6B not seen in Figure 2 Supp 1?).Several mechanisms could account for the lack of WhyD activity, including a proteolytic cleavage, releasing the catalytic domain away from nascent peptidoglycan or the lack of a co-factor. I certainly agree that WhyD is present throughout growth, but it is possible that the amount of protein decreases in autolytic phase. The authors should be more cautious and/or quantify the relative amount of WhyD throughout growth (using densitometry). "WhyD levels remain constant throughout growth and under autolytic conditions" is a bit assertive given the experimental evidence shown.

We agree with the reviewer that WhyD activity is likely to be (somehow) reduced during autolytic conditions. The molecular basis for this reduction in activity remains unclear and is an active area of investigation. We now discuss possible mechanisms in the Discussion. We have also added molecular weight markers to the anti-WhyD immunoblots to provide better clarity.

Figure 2 —figure supplement 1 shows full-length WhyD protein levels throughout growth. The other bands in this immunoblot are cross-reacting proteins detected by our antiserum as they are also detected in our lysates derived from cells deleted for *whyD.*

Figure 6 —figure supplement 1 shows immunoblots to detect full-length GFP-WhyD, which is not very stable. We think that the difference in the non-specific bands between

Figure 2 —figure supplement 1 and Figure 6 —figure supplement 1 is likely due to the use of different transfer methods. We had problems with variability when blotting for GFP-WhyD using our standard semi-dry transfer technique that we used for detecting untagged WhyD in Figure 2 —figure supplement 1. We therefore used wet transfer for immunoblots to detect GFP-WhyD. We suspect the change in transfer method results in differences in the non-specific proteins detected, perhaps due to differing efficiencies of transfer between methods. We now indicate in the methods when we used each transfer technique to provide clarity for the reader.

Finally, Figure 2- supplement 1 shows that the steady-state levels of full-length WhyD protein levels remain unchanged during growth and autolysis, strongly suggesting that the protein is not subjected to enhanced proteolysis during autolysis.

Reviewer #2 (Recommendations for the authors):I really like the paper and I think there should not be much work to be done to get it ready for publication. I have my comments and suggestion in order as it shows in the manuscript:Line 7: "Penicillin and related antibiotics disrupt cell wall synthesis in bacteria and induce lysis by misactivating cell wall hydrolases called autolysins"This sentence should be re-written. As it is, it misleads non-experts to think that Penicillin stops cell wall synthesis BY directly activating hydrolases, resulting in lysis. That's not the broad mechanism of Penicillin. The central problem with the sentence might be "misactivating". I understand that the conclusion of the paper solidifies the idea that, in the absence of proper cell wall synthesis, hydrolase misactivation results in lysis and this is important in the context of some antibiotics. But again, you should be careful not to confuse the general public target of this journal. This also re-appears in line 97.

We have modified the sentence (line 48-49) to address this concern. It now reads:

“Penicillin and related antibiotics disrupt cell wall synthesis in bacteria causing the downstream misactivation of cell wall hydrolases called autolysins to induce cell lysis.”

Line 51: "We recently discovered that penicillin treatment of Sp causes a dramatic shift in surface polymer biogenesis in which cell wall-anchored teichoic acids (WTAs) increase in abundance at the expense of lipid-linked lipoteichoic acids".You can simplify "lipid-linked lipoteichoic acids" by "lipoteichoic acids" as "lipo" already implies "lipid-linked". For consistency, you can ass "(LTA)" at the end to make the contrast with "WTA".

Thank you. It now reads: " at the expense of lipid-linked teichoic acids (LTAs)"

Line 54: "Because LytA binds to these polymers,…"What polymer? PG? WTA? LTA? Please be specific.

Thank you: It now reads "Because LytA binds to both species of teichoic acids…”

Line 93: "They also play important roles in cleaving shared cell wall material connecting daughter cells during cytokinesis (Vollmer et al., 2008)".I suggest adding this interesting and important paper as well: https://pubmed.ncbi.nlm.nih.gov/29458657/

Added.

Also, as far as I know, the most comprehensive work on the global deletion of all hydrolases in Bacillus and how the majority of them play a role in daughter cell separation. This same reference can be used in lines 107-110: https://www.biorxiv.org/content/10.1101/2021.02.18.431929v1

Added.

I also like the following: https://pubmed.ncbi.nlm.nih.gov/12399477/

Added.

Line 161: "Furthermore, LytA production in cells lacking WhyD during growth in liquid medium caused premature lysis in late exponential phase (Figure 1C)."Why would this happen only in the late exponential/stationary phase? Or this is only detected in bulk growth at this stage but it's consistently happening throughout all phases?

Autolysis is a multifactorial process. It not only requires an increase in WTAs but also a reduction in LTAs that sequester LytA away from the cell wall. The reduction in LTAs is mediated by degradation of the LTA synthase TacL in early stationary phase. Upon its degradation, LTA levels go down due to their shedding into the medium. We suspect these factors contribute to the timing of lysis in the ∆*whyD* mutant.

Figure 1D: why the concentration of purified WhyD needs to be so high (1mg/ml)? If a titration was done, could be interesting to include the data and a sentence.

We thank the reviewer for catching this. The concentration was 1µg/ml for rLytA and 10 µg/ml for WhyD^CT^ (not 1mg/ml). We have modified the text accordingly.

Line 204: "To test this possibility, we purified the C- terminal GlpQ domain of WhyD (WhyDCT) and monitored its ability to release WTAs from purified sacculi (Figure 3B)".Make sure to clarify that the reason to purify the GlpQ sub-domain alone is because of the difficulty to purify it together with the transmembrane region.

We have modified the text (lines 207-210) to clarify this. The text now reads:

“To test this possibility and to facilitate purification, we expressed and purified the soluble C-terminal GlpQ domain of WhyD (WhyD^CT^_;_ and Figure 3 —figure supplement 1), and monitored its ability to release WTAs from purified sacculi (Figure 3B).”

Figures 4 and 5: I fail to understand what the GFP, HADA, and sBADA are doing there. And why sometimes do you use one or another probe? There is nothing about it in the text. Also, I strongly recommend including sample N in every plot. If possible, please change distributions to superplots where we can visualize the distributions of each biological replicate.

Cytoplasmic GFP was used to visualize cell size and create meshes for quantification. FDAAs were used to monitor cell cycle progression and identify recently synthesized PG and newly synthesized (nascent) PG. We have added this information to the Materials and methods to clarify why each probe was used.

We thank the reviewers for requesting these data. We have included information about replicates (N number) in Materials and methods.

Figure 4: I noticed from the images that the major difference between different genetic backgrounds may come from cells in duplets/short chains compared to long chains. It would be interesting to point out if that's true (cell wall, WTA, and LTA composition may vary with cell-cell association). It should be easy to perform this analysis and correlate chain length and cell length/area/width. This could bring new insights on whether the effect seen in the previous experiments could come majorly from short chains (enriched in late exponential and stationary phases). Also, do short chains grow faster / slower / the same as long chains? Are they more sensitive to antibiotics?

The cropped regions presented in Figure 4 gives the impression that the ∆*whyD* mutant forms longer chains of cells. However, viewing larger fields of cells, the difference between these backgrounds is quite modest. We have included larger fields in a supplemental figure to help clarify this point (Figure 4 —figure supplement 1).

Figure 5: if authors want to imply cells are smaller due to elongation defects, wouldn't be much easier to simply perform single-cell tracking from time-lapses instead of indirectly staining the cell wall?

Our analysis is similar to the approach taken by others in the field and we think it is sufficient. We acknowledge that single-cell tracking is more direct, but given the small size of *Sp* cells and their fastidious growth, these measurements are not trivial with this organism.

Figure 6: Even though the rLytA-Alexa data is very compeling, I am not convinced GFP-WhyD is recruited to midcell. It could be simply that cells have double the membrane at midcell. Authors could make their point stronger by staining membranes and quantitating membrane:GFP-WhyD ratios to prove there is indeed accumulation of WhyD compared to the cell body.

We performed line scans of the GFP-WhyD signal and the signal was more than double the signal at the poles. More importantly, we think our description of our GFP-WhyD fusion and its localization were sufficiently cautious such that we kept the text as is.

Figure 7-Sup1-2-3: Please, make sure to show another panel with the background signal when LytA does not bind cells. It's very surprising that there is no signal at all. Even inespecific signal.

Done.

Reviewer #3 (Recommendations for the authors):I have no major objection and only some suggestions that could improve the manuscript.– Page 6, line 153 : WhyD is assumed to be essential but some deletions can be obtained in WT cells. Does suppressive mutations occurs only in lytA or other genes encoding hydrolases? Whole genome sequencing of some strains would help to strengthen authors hypothesis that WhyD modulates WTA localization and thus localization of hydrolases involved in cell elongation and final separation.

The small number of Tn insertions detected in the *whyD* gene in the Tn-Seq profile of WT cells in Figure 2A are likely due to spurious PCR amplification. It is relatively rare to have a gene completely devoid of insertions in a Tn-Seq profile.

That said, we thank this reviewer for this interesting point. When we started this project, we were also interested in investigating whether deletion of other choline-binding hydrolases (in addition to LytA) would suppress the lethality of *whyD* inactivation. As the reviewer points out, the increase of WTAs seen in cells lacking *whyD* should lead to higher levels of other choline-binding PG hydrolases associated with WTAs and therefore adjacent to the cell wall.

To test this possibility, we attempted to inactivate *whyD* in different strain backgrounds lacking one or several choline-binding PG hydrolases (single mutants: D*lytB*, D*lytC*, D*cbpD*; or the triple mutant lacking all three). In all cases, we found *whyD* to be essential. Transforming D*whyD* into these strains yielded >500-fold fewer transformants compared to transformation in the ∆*lytA* mutant. In hindsight, these results are not surprising given that only LytA has been implicated in antibiotic-induced and growth-phase-dependent bacteriolysis, both phenomena that are impacted by the levels of WTAs in the cell wall.

– To construct the ∆whyD∆lytA, I guess the author deleted whyD in the ∆lytA background? Was transformation efficiency similar to WT cells transformed with a control DNA? If not, it could be worth identifying the additional suppressive mutations to establish a genetic link with other hydrolases or factors promoting cell elongation.

The reviewer is correct: the ∆*whyD* allele was transformed into a ∆*lytA* mutant strain. The transformation efficiency of the ∆*whyD* allele was comparable to an unrelated mutant using matched amounts of input DNA.

– To be consistent with their previous binding, does whyD overexpression suppress at least the essentially of tacL (that requires a mutation in cozE)?

This is an interesting point and something that we attempted in our initial characterization of *whyD*. However, we found that overexpression of *whyD* dramatically reduces competence. The reviewer is also correct that *tacL* inactivation in the *lytA* background also requires a mutation in the essential protein *cozE*. Due to these technical difficulties we did not pursue this further.

– Regarding the link between lytA and whyD. Did the author try to evaluate the essentiality of whyD in other hydrolases mutant that also bind choline? For instance, the hydrolase CbpD, that is expressed during competence in the early exponential phase, also possesses choline-binding domain. Are the same observations made in a ∆cbpD mutant?

See comments above.

– Page 8, lines 206-208: WhyD clearly releases WTA from PG sacculi, but a significant amount remains linked. Did the authors try to make longer incubation with more WhyD? it is possible that a fraction is resistant to WhyD hydrolysis? If yes, can the author comment on that? It could be due other modifications of PG (acetylation) or WTA (A-alanylation). It could make sense with the suppressive mutation detected upon deletion of whyD in WT cells (see my two first comments).

Figure 3B shows that most (virtually all) WTAs are released from PG sacculi by purified WhyD^CT^. Previous studies found that cleavage of PG sacculi from *B. subtilis* by cell wall hydrolases is less efficient than gram-negative sacculi presumably due to the thicker cell wall in this non-native context (PMID: 20159959, 34871030, and 31808740). It is therefore difficult to interpret the absence of complete cleavage. That said, WhyD^CT^ seems to be quite active.

– Page 11 and Figure 6: From the phase contrast image, it seems that cells chain. Does quantification confirm this visual impression? If yes, it would mean that the lower ectopic expression of GFP-WhyD (compared to whyD expression in WT cells (Fig6-Sup1B)) would also impact the activity of LytB? Does addition of purified recombinant LytB (as described in Rico-Lastress et al., Sci. Rep., 2015) abolish cell chaining? If yes, it would provide a direct link between WhyD and LytB.

Cells with reduced levels of WhyD chain a little more than wild-type but the effect is modest. It is possible that the increased WTAs results in mislocalization of LytB away from the septum since is also a choline binding protein. However, recent data from the Grangeasse group indicates that LytB localization also requires an interaction with StkP, which complicates the picture. We have not tried adding LytB to cells lacking WhyD but it is an experiment worth trying in the future. The cropped regions presented in Figure 4 gives the impression that the ∆*whyD* mutant forms longer chains of cells. However, viewing larger fields of cells, the difference between these backgrounds is quite modest. We have included larger fields in a supplemental figure (Figure 4 —figure supplement 1) to clarify this point.

– I would be great to analyze the dynamic of WhyD localization in the course of the cell cycle. Maybe the fluorescence signal is not compatible with time-lapse microscopy but sorting and plotting cells as a function of the different cell division stages may help to address this point and notably if WhyD localizes early or late at the division septum?

We would have loved to do the requested experiment. Unfortunately, the WhyD-GFP signal is quite faint and bleaches after one or two exposures. Instead, we used demographs to sort cells based on size (Figure 6).

– Page 12, line 309: there is actually no rLytA* signal co-localizing with BADA in elongating cells in the Figure!

We thank the reviewer for highlighting this point. In fact, our central conclusion is that the vast majority of nascent PG synthesis (sBADA labeling) does not co-localize with WTAs (rLytA* labeling). Thus, WhyD is likely to be removing WTAs from most newly synthesized PG.

– Figure 7B: White head arrow could be colored in blue to be consistent with the color code of Figure 7A.

Done (thank you).

– Page 13, line 322: Figure 7D instead of Fig7C

Text has been amended, thank you for catching this.

– Figure 2-FigSup1 and others: What is the protein detected as the LC?

The Loading Control is a cross-reacting protein recognized by our anti-WhyD antiserum.

– Figure 3: an amino acid sequence alignment would be helpful to appreciate the homology with GlpQ phosphodiesterase domain.

We thank the reviewer for requesting this. We now include an alignment in Figure 3 —figure supplement 1.